# RT-Lynx: Putting GEMM Sparsity in the Right Place for Diffusion Models

Xing Cong[1] Hanlin Tang[†2] Kan Liu[2] Lan Tao[2] Lin Qu[2] Chenhao Xie[‡1]

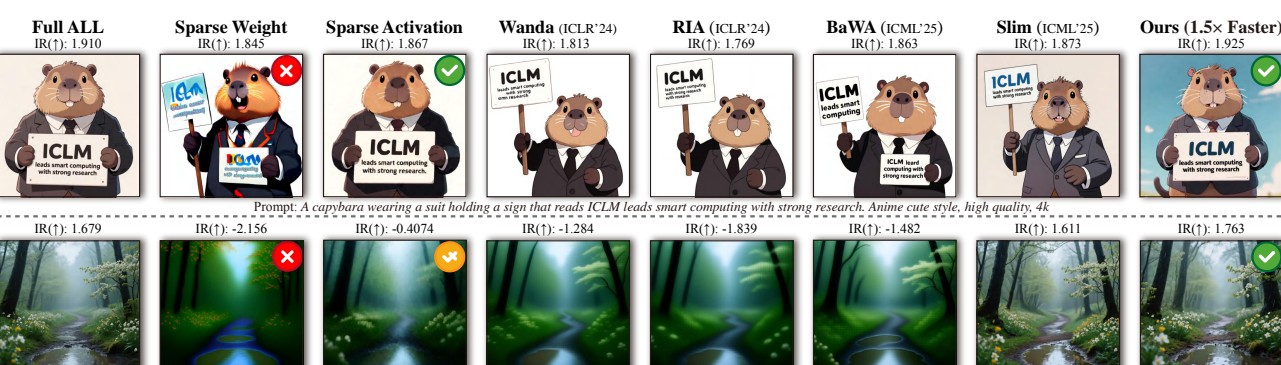

**Figure 1.** Comparison of sparsification strategies on Qwen-Image (Wu et al., 2025). The figure illustrates the visual impact of different methods on generation quality. Both weight sparsity and activation sparsity apply Top-$K$ selection to enforce 2:4 semi-structured sparsity. IR(↑) reports the Image Reward (Xu et al., 2023) for each image.

## Abstract

Diffusion Transformers (DiT) achieve strong performance in image generation but incur substantial inference costs. While prior work has reduced this cost via quantization and distillation, semi-structured sparsity—which can nearly halve FLOPs—remains underexplored. A key reason is that most existing approaches focus on weight sparsification, and pruning 50% of the weights can remove critical model capacity and degrade generation quality. Our study, however, shows that DiT activations are intrinsically sparse and significantly more robust to N:M semi-structured sparsification than weights. Motivated by this observation, we advocate a paradigm shift from weight sparsification to activation sparsification. We propose RT-Lynx, which applies N:M sparsification to activations and incorporates error-compensation techniques to mitigate accuracy loss. We further implement highly optimized CUDA kernels tailored to this setting, achieving up to a 1.55× speedup on average in linear layers. Extensive experiments across multiple diffusion models demonstrate that our method preserves the generation quality of the original models while substantially accelerating inference.

## 1. Introduction

Diffusion models have recently achieved remarkable progress in high-quality image generation (Ho et al., 2020). By introducing Transformer-style global modeling into diffusion, Diffusion Transformers (DiT) demonstrate superior generation quality, diversity, and scalability(Peebles & Xie, 2023; Meng et al., 2021; Ruiz et al., 2023; Zhang et al., 2023; 2025a; Zhuo et al., 2025), and have become a core paradigm for high-resolution, high-fidelity synthesis(Bai et al., 2024; Esser et al., 2021; 2024; Yang et al., 2025; Feng et al., 2025). Despite these advantages, DiT faces severe inference efficiency bottlenecks in practice due to the compute-intensive nature of each step and the necessity of tens of iterative diffusion steps, which together amplify latency and energy overhead. Addressing DiT's inference cost without degrading generation quality has therefore become a critical challenge.

In large language models (LLMs), sparsification (also referred to as *pruning*)(Frantar & Alistarh, 2023; Sun et al., 2023; Zhang et al., 2024a; Liu et al., 2025g; Mozaffari et al., 2025) has been established as an effective technique for inference acceleration. In particular, N:M semi-structured sparsity ("sparsity" throughout this work unless otherwise specified) has been extensively studied due to its favorable accuracy–performance trade-off and native hardware support (Fang et al., 2024), as exemplified by representative

---
[†] Project leader. [‡] Corresponding author. [1]Beihang University, School of Computer Science, Beijing, China [2]Independent Researcher, Beijing, China. Correspondence to: Xing Cong <congxing@buaa.edu.cn>, Chenhao Xie <xiechenhao@buaa.edu.cn>.

*Proceedings of the 43rd International Conference on Machine Learning*, Seoul, South Korea. PMLR 306, 2026. Copyright 2026 by the author(s).

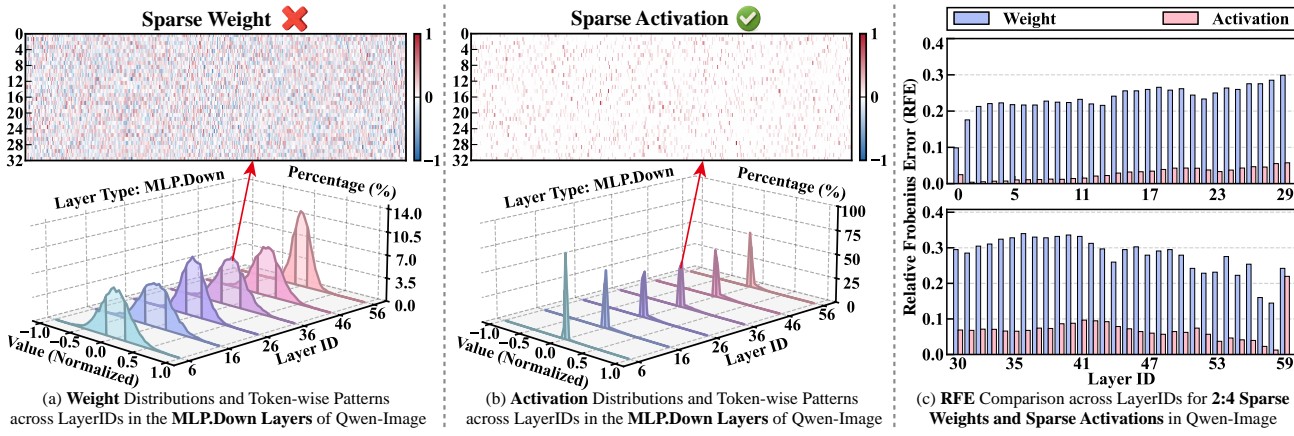

(a) **Weight** Distributions and Token-wise Patterns across LayerIDs in the **MLP.Down Layers** of Qwen-Image

(b) **Activation** Distributions and Token-wise Patterns across LayerIDs in the **MLP.Down Layers** of Qwen-Image

(c) **RFE** Comparison across LayerIDs for **2:4 Sparse Weights and Sparse Activations** in Qwen-Image

*Figure 2.* Rethinking sparsity in DiT through distributions and induced errors. (a)–(b) compare weights and activations in mlp.down layers of Qwen-Image: the lower parts show normalized value distributions across six depths (with [-1,1] divided into 50 bins and lighter regions indicating values below the median), while the upper parts visualize row-wise heatmaps from the 36th layer. The lighter colors denote values closer to zero. (c) reports the relative errors induced by Top-$K$ 2:4 sparsification across layer depths.

methods such as SparseGPT (Frantar & Alistarh, 2023) and Wanda (Sun et al., 2023). Despite extensive academic progress, sparsification has seen limited adoption in production systems. One critical challenge is the noticeable accuracy degradation: prior weight sparsification methods such as SparseGPT and Wanda report accuracy drops exceeding 3% when pruning 50% of the weights (Frantar & Alistarh, 2023; Sun et al., 2023). Our study reaches similar conclusions for DiT models. As shown in Figure 1, weight sparsification severely compromises the original model's image generation capability. Moreover, we observe the weight statistics (Figure 2(a)) exhibit an unclear semi-structured sparsity pattern, hindering the realization of effective sparsification for DiT.

In contrast to the weights, we find that token-wise activations are intrinsically sparse. Figure 2(b) shows that, within each token, only a small subset of channels is substantially activated. Consequently, enforcing sparsity on activations induces significantly smaller output error (Figure 2(c)) and yields notably better visual quality than weight sparsification (see Figure 1). These results motivate a key paradigm shift: **instead of enforcing semi-structured sparsity on weights, we advocate activation sparsification, leveraging the inherent sparsity of per-token activations**.

Motivated by these findings, we propose RT-Lynx, an end-to-end solution for DiT sparsification. Notice that naive activation pruning can still incur non-negligible quality degradation, so we develop several error-compensation strategies (Section 4) to mitigate the side effects of sparsification. Meanwhile, we want to emphasize that inference acceleration should serve as an important criterion for sparsity and the core goal of sparsification. Following that proposal, we design a unified CUDA inference pipeline that fuses online N:M sparsification with sparse Tensor Core execution, enabling low-overhead and effective end-to-end acceleration. Our highly optimized CUDA kernel achieves up to 1.88×

Sparse GEMM speedup and 1.55× linear-layer speedup. To the best of our knowledge, this is the first work to achieve high-speedup lossless N:M sparsification for DiT models.

Our main contributions are summarized as follows:

- We identify a fundamental shift in the sparsification paradigm for DiT: activations are substantially more robust to semi-structured sparsification than weights.
- We propose RT-Lynx, which combines norm-based compensation and LoRA adaptation to fully recover model performance after sparsification.
- We design a plug-and-play sparse inference pipeline that fuses online N:M sparsification with sparse Tensor Core execution to deliver practical end-to-end speedups.
- Extensive evaluations on mainstream DiT models demonstrate consistent acceleration with negligible quality degradation.

## 2. Preliminary

### 2.1. N:M Sparsity

Semi-structured sparsity (Figure 3) enforces fixed local N:M patterns, retaining only N nonzero elements in each of M numbers(Bai & Li, 2023; Lin et al., 2023). This regularity enables efficient hardware decoding and motivates NVIDIA and other vendors to introduce Sparse Tensor Cores (SpTC) for such computations, providing up to 2× theoretical acceleration (see Appendix A).

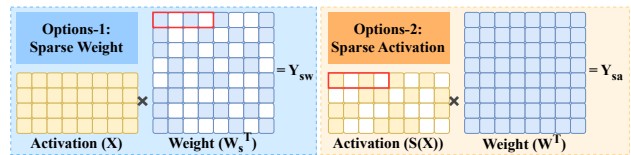

*Figure 3.* Weight Sparsity vs. Activation Sparsity under 2:4 semi-structured patterns. white blocks denote discarded elements.

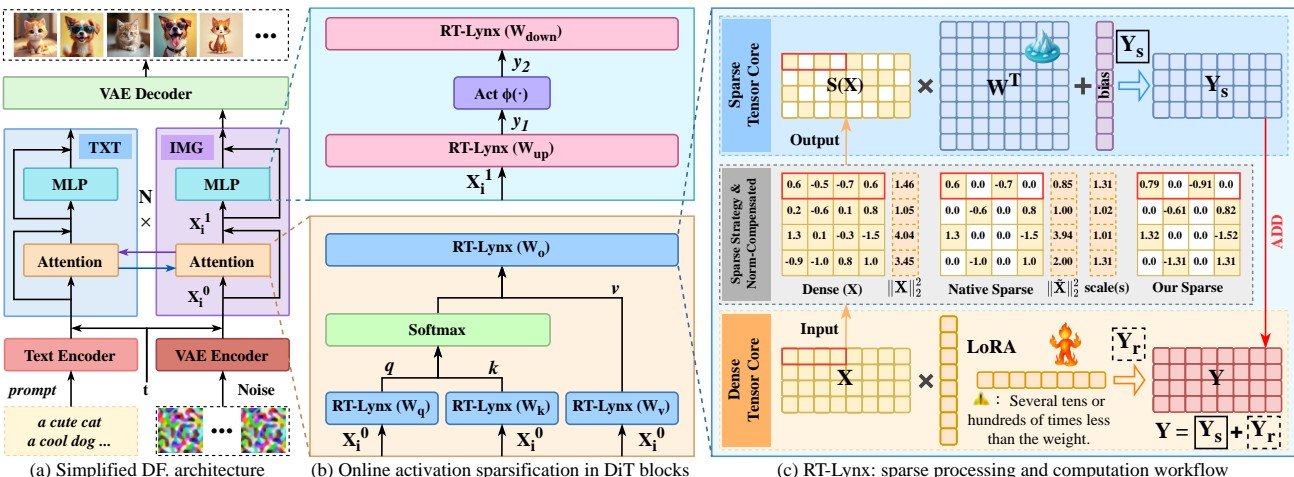

(a) Simplified DF. architecture     (b) Online activation sparsification in DiT blocks     (c) RT-Lynx: sparse processing and computation workflow

*Figure 4.* (a) illustrates the abstract architecture of the DiT model, highlighting the Transformer modules selected for sparsity analysis. (b) indicates the locations where activation sparsity is applied, primarily including the QKV projections before attention computation and the Up and Down mappings in the MLP. (c) depicts the overall pipeline of online activation sparsification with LoRA compensation, where the backbone weights are kept frozen, and only the low-rank LoRA matrices are efficiently fine-tuned.

## 2.2. Weight Sparsity and Activation Sparsity

A Linear layer performs a matrix multiplication

$$\mathbf{Y} = \mathbf{X} \cdot \mathbf{W}^T \tag{1}$$

where $\mathbf{X} \in \mathbb{R}^{N \times D_{\text{in}}}$ is the input, $\mathbf{W}^T \in \mathbb{R}^{D_{\text{in}} \times D_{\text{out}}}$ is the weight matrix, and $\mathbf{Y} \in \mathbb{R}^{N \times D_{\text{out}}}$ is the output. With an N:M sparsity pattern, this computation admits two feasible paths (Figure 3). The first applies N:M sparsity to the weights, yielding a static sparse matrix

$$\mathbf{Y} = \mathbf{X} \cdot \mathbf{W}_s^T \tag{2}$$

where $\mathbf{W}_s^T$ follows an N:M structure and remains fixed at inference, enabling direct use of hardware sparse kernels. The second applies N:M sparsity to the activations,

$$\mathbf{Y} = S(\mathbf{X}) \cdot \mathbf{W}^T \tag{3}$$

where $S(\cdot)$ enforces an N:M pattern on $\mathbf{X}$ at runtime. The former is static and model-dependent, while the latter is dynamic and input-adaptive.

## 3. Motivation

Previous studies on LLM sparsification (Frantar & Alistarh, 2023; Sun et al., 2023; Zhang et al., 2024a; Liu et al., 2025g; Mozaffari et al., 2025) have consistently reported a non-negligible accuracy degradation under the 2:4 sparsity pattern, and our results reach a similar conclusion. On Qwen-Image (see Figure 1), we compare native activation sparsity, conventional weight sparsity, and recent state-of-the-art weight sparsification methods, and observe that all weight-sparse models perform markedly worse than the dense baseline—even with the most advanced strategies—whereas native activation sparsity exhibits significantly more promising potential. To further characterize this phenomenon, we conduct a comparative study of weight sparsity and activation sparsity under the 2:4 patterns on Qwen-Image (Wu

et al., 2025). These results motivate us to investigate a new sparsification paradigm.

### 3.1. Weights are not Intrinsically Sparse

Although it is widely acknowledged that a certain proportion of trivial elements in model weights can be safely pruned, our analysis reveals that model weights are not inherently trained to be sparse. As shown in Figure 2(a), individual weight elements follow a quasi-Gaussian distribution and are broadly, almost randomly, spread across the normalized range, indicating the absence of intrinsic structured patterns such as 2:4 sparsity. This stochastic distribution implies that weights do not naturally align with structured sparsity constraints, and enforcing such a pattern inevitably removes salient parameters. Consistently, reconstruction errors measured by RFE (Figure 2(c)) demonstrate that weight sparsification incurs substantial and highly sparsification errors across all layers.

### 3.2. Activations are More Sparse Due to Superposition

Previous work (Cunningham et al., 2023) on LLM interpretability shows that Transformers exhibit a token-level superposition mechanism, where each token, associated with a specific concept, activates only a small subset of neurons in FFNs. Our analysis confirms this behavior in Figure 2(b): activations concentrate sharply near zero, with only about $5\% \sim 10\%$ of neurons being active. This intrinsic sparsity stands in sharp contrast to the quasi-Gaussian and dense distribution of model weights, and implies that structured constraints mainly eliminate near-zero activation values rather than salient information. Consequently, this highly sparse pattern makes activations a much better choice for inducing model sparsity.

# 4. Methodology

Although activation sparsity consistently achieves higher generation quality than weight sparsity, it still introduces mild blurring without careful tuning (see Figure 1). Moreover, activation sparsification is performed online, which can incur prohibitive computational overhead (exceeding 40% of the total runtime without optimization as shown in Table 5). These issues collectively hinder the realization of a truly lossless model sparsification pipeline. To address these challenges, we introduce **RT-Lynx**, an end-to-end framework that integrates DiT models with sparse GEMM, delivering strong quality guarantees alongside substantial performance improvements.

---

**Algorithm 1** RT-Lynx

---

1: **Input:** activation $\mathbf{X} \in \mathbb{R}^{N \times D_{\text{in}}}$, weight $\mathbf{W} \in \mathbb{R}^{D_{\text{out}} \times D_{\text{in}}}$, LoRA $\mathbf{L}_A \in \mathbb{R}^{D_{\text{out}} \times R}$, $\mathbf{L}_B \in \mathbb{R}^{R \times D_{\text{in}}}$, 2:4 structured Top-$k$ operator $\text{TOPK}(\cdot)$, $\epsilon$.

2: **Output:** $\mathbf{Y} \in \mathbb{R}^{N \times D_{\text{out}}}$.

3: $\tilde{\mathbf{X}} \leftarrow \text{TOPK}(\mathbf{X})$        // keep 2 per group

4: $s \leftarrow \sqrt{\frac{\|\mathbf{X}\|_2^2}{\|\tilde{\mathbf{X}}\|_2^2 + \epsilon}}, \quad \mathbf{S}(\mathbf{X}) \leftarrow s \cdot \tilde{\mathbf{X}}$

5: $\mathbf{Y}_s \leftarrow \mathbf{S}(\mathbf{X})\mathbf{W}^\top, \quad \mathbf{Y}_r \leftarrow \mathbf{X}(\mathbf{L}_A\mathbf{L}_B)^\top$

6: $\mathbf{Y} \leftarrow \mathbf{Y}_s + \mathbf{Y}_r$

7: **return** $\mathbf{Y}$

---

## 4.1. Norm-Compensated Sparsification

Pruning elements directly from activations inevitably reduces their overall norm. To mitigate this effect, we propose a norm-compensated activation sparsification scheme that explicitly preserves the $l_2$-norm of the original activation. The key idea is to rescale the sparse activation so that its magnitude matches its dense counterpart.

Concretely, consider an activation $\mathbf{X} \in \mathbb{R}^{1 \times 4}$ under a 2:4 sparsity constraint. A Top-2 operator retains the two largest entries and produces a sparse vector $\tilde{\mathbf{X}} = \{x_i \mid i \in \mathcal{I}\}$. The final sparse activation is defined as

$$S(\mathbf{X}) = s \cdot \tilde{\mathbf{X}}, \quad s = \sqrt{\frac{\|\mathbf{X}\|_2^2}{\|\tilde{\mathbf{X}}\|_2^2 + \epsilon}}. \qquad (4)$$

Here, $\epsilon = 1e{-}8$ ensures numerical stability. This formulation restores the magnitude of $\tilde{\mathbf{X}}$ to that of $\mathbf{X}$, effectively eliminating norm attenuation induced by sparsification while introducing only negligible computational overhead.

## 4.2. LoRA Adaptation and Fine-Tuning

Despite most activation elements being close to zero, they still encode fine-grained details in the generated images. Our empirical results show that these low-magnitude activations mainly affect high-frequency visual details, such as hair, edges, and textures; directly removing them can therefore introduce blurring or local artifacts. To recover this residual information, we introduce a lightweight LoRA branch to compensate for the sparsification error. The rationale is that such high-frequency residuals account for only a small fraction of the overall image information, and thus can be effectively modeled by a low-rank branch. More specifically, denote the norm-compensated activation as $S(\mathbf{X})$, the final result is computed as:

$$\mathbf{Y} = \mathbf{Y}_s + \mathbf{Y}_r = S(\mathbf{X}) \cdot \mathbf{W}^\top + \mathbf{X} \cdot (\mathbf{L}_A\mathbf{L}_B)^\top \qquad (5)$$

where $\mathbf{Y}_s \in \mathbb{R}^{N \times D_{\text{out}}}$ denotes the output computed from sparse activations, and $\mathbf{Y}_r \in \mathbb{R}^{N \times D_{\text{out}}}$ represents the compensation residual. Here, $\mathbf{L}_A \in \mathbb{R}^{D_{\text{out}} \times R}$ and $\mathbf{L}_B \in \mathbb{R}^{R \times D_{\text{in}}}$ are the LoRA matrices. We set $R = 64$ to balance accuracy and inference overhead. In this case, the pruned activations are further recovered by the low-rank branch. The final loss of the Lora training is to minimize the output discrepancy between the original output and the sparsed one, which can be formalized as:

$$Loss = \left\| \mathbf{X} \cdot \mathbf{W}^\top - \left( S(\mathbf{X}) \cdot \mathbf{W}^\top + \mathbf{X} \cdot (\mathbf{L}_A\mathbf{L}_B)^\top \right) \right\|^2.$$

Our results show that the training converges within 2k steps. The overall inference pipeline is shown in Algorithm 1. Further details are provided in Appendix D.

## 4.3. Selective Layer Skipping in Single-stream DiT

Although the proposed sparsification strategy with LoRA adaptation is effective for double-stream DiT architectures (e.g., Qwen-Image), we observed that on single-stream paths, a noticeable performance gap persists that cannot be fully addressed by the LoRA branch. Therefore, we skipped certain linear layers on Z-Image and FLUX. On Z-Image, we skipped the attn.o_proj and mlp.up layers in the single-stream paths; On FLUX, we skipped the attn.o_proj and mlp.down layers in the single-stream paths. Full details about this layer choice can be found in Appendix E.

## 4.4. CUDA Kernel Optimization

While shifting from weight sparsity to activation sparsity improves robustness and better preserves generation quality, achieving end-to-end inference speedups remains challenging due to two system-level inefficiencies: (1) Online activation sparsification incurs high overhead, which can occupy nearly 40% of runtime, reaching up to 59% in practice (Table 5 and Figure 5(a)). (2) The dense LoRA branch is typically executed as an isolated path with intermediate materialization, introducing avoidable memory traffic and synchronization.

To overcome these inefficiencies, we design an online sparse execution framework with two optimizations:

- We fuse the entire online sparsification pipeline (Figure 5(b))—pattern determination, Top-$K$ selection, and

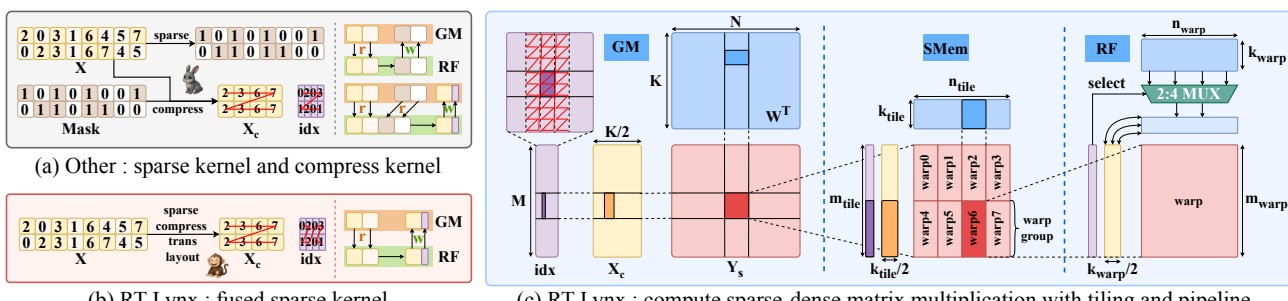

(a) Other : sparse kernel and compress kernel

(b) RT-Lynx : fused sparse kernel

(c) RT-Lynx : compute sparse-dense matrix multiplication with tiling and pipeline

*Figure 5.* The figure compares conventional and proposed sparse execution pipelines: (a) existing N:M sparse methods separate pruning, formatting, and computation; (b) the proposed online framework fuses these steps into a unified execution flow; and (c) illustrates the tiled and pipelined sparse matrix multiply–accumulate on SpTC.

*Table 1.* Quantitative image quality comparison of different sparsity strategies on Qwen-Image. Bold denotes the best strategy within each row group under the corresponding metric. Besides Full, the table contains two row groups: naive 2:4 sparsity applied to weights and activations, and a comparison between our method and SOTA weight sparsification methods. More results are provided in Appendix G.1

| Method | MJHQ | | | | sDCI | | | |
|---|---|---|---|---|---|---|---|---|
| | FID($\downarrow$) | IR($\uparrow$) | C.SCR($\uparrow$) | C.IQA($\uparrow$) | FID($\downarrow$) | IR($\uparrow$) | C.SCR($\uparrow$) | C.IQA($\uparrow$) |
| Full | 21.98 | 1.219 | 27.12 | 0.9317 | 31.15 | 1.172 | 26.49 | 0.9492 |
| Sparse Weight | 51.63 | -0.1605 | 24.22 | 0.5783 | 66.91 | -0.2159 | 24.93 | 0.5792 |
| Sparse Activation | **35.85** | **0.5994** | **26.28** | **0.7473** | **48.59** | **0.4724** | **26.13** | **0.7550** |
| Wanda (ICLR'24) | 40.81 | 0.5356 | 26.06 | 0.7344 | 55.61 | 0.3245 | 26.01 | 0.7268 |
| RIA (ICLR'24) | 43.02 | 0.4627 | 25.90 | 0.7150 | 58.90 | 0.2325 | 25.90 | 0.7124 |
| BaWA (ICML'25) | 39.68 | 0.5888 | 26.15 | 0.7660 | 54.54 | 0.3761 | 26.15 | 0.7692 |
| Slim (ICML'25) | 22.25 | 1.278 | 27.18 | 0.9242 | 29.26 | 1.217 | 26.23 | 0.9350 |
| RT-Lynx (**Ours**) | **21.25** | **1.304** | **27.33** | **0.9263** | **25.78** | **1.226** | **26.61** | **0.9366** |

compression—into a single CUDA execution path, generating 2:4 structured activations directly in SpTC-compatible layouts at the register level, making sparsification a lightweight in-kernel procedure. On this basis, the sparse GEMM kernel employs a streamK-style, block-parallel pipeline that effectively exploits the bandwidth–latency hierarchy across storage tiers, maximizing sparse compute efficiency.

• We construct an integrated workflow that interleaves sparse computation with dense LoRA execution: sparse GEMM produces $\mathbf{Y}_s$, while the LoRA branch computes $\mathbf{Y}_r$ on dense tensor cores; $\mathbf{Y}_s$ is then accumulated into $\mathbf{Y}_r$ on-chip to form $\mathbf{Y}$, eliminating LoRA-intermediate materialization and reducing synchronization overhead.

## 5. Experiments

### 5.1. Setups

**Models**. We evaluate the method on three representative DiT architectures with four configurations: Qwen-Image (Wu et al., 2025) (initial and 2512 versions), FLUX.1 (Labs et al., 2025), and Z-Image (Team, 2025). These models span diverse scales and structures, enabling the evaluation of activation sparsity and the online execution framework across heterogeneous settings. Detailed configurations are given in Appendix F.1.

**Datasets**. We randomly sample 20k prompts from our collected user requests and use Qwen-Image to generate the corresponding images. The resulting prompt–image pairs are used as training samples. To evaluate the effectiveness of our method, we follow the standard text-to-image evaluation protocol and draw prompts from MJHQ-30K (Li et al., 2024a) and sDCI (Urbanek et al., 2024; Li et al., 2024b), sampling **5,000 prompts** from each dataset. Detailed dataset statistics are provided in Appendix F.2.

**Baselines**. To contrast activation and weight sparsity and evaluate the LoRA adaptation, we compare against several SOTA weight-sparsification methods that require no large-scale retraining, including Wanda (Sun et al., 2023), RIA (Zhang et al., 2024a), BaWA (Liu et al., 2025g), and Slim (Mozaffari et al., 2025) (details in Appendix F.3). We further benchmark system efficiency against PyTorch-GEMM (Paszke et al., 2019), PyTorch-SpMM (Cai et al., 2024), cuSPARSElt (NVIDIA Corporation, 2025), and CUTLASS (NVIDIA Corporation, 2026). Under N:M sparsity, all methods adopt the same 2:4 pattern.

**Metrics**. We evaluate generation quality under FP16 using FID (Heusel et al., 2017; Parmar et al., 2022), Image Reward (IR) (Xu et al., 2023), and CLIP-IQA (C.IQA) (Wang et al., 2023) with CLIP-Score (C.SCR) (Hessel et al., 2021), covering distributional fidelity, human preference, percep-

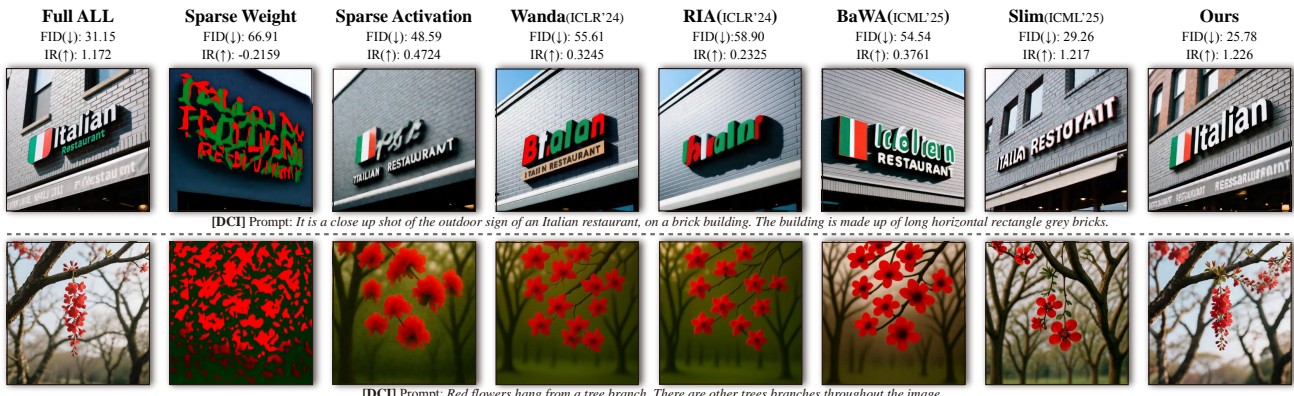

| **Full ALL** | **Sparse Weight** | **Sparse Activation** | **Wanda**(ICLR'24) | **RIA**(ICLR'24) | **BaWA**(ICML'25) | **Slim**(ICML'25) | **Ours** |
|---|---|---|---|---|---|---|---|
| FID(↓): 31.15 | FID(↓): 66.91 | FID(↓): 48.59 | FID(↓): 55.61 | FID(↓):58.90 | FID(↓): 54.54 | FID(↓): 29.26 | FID(↓): 25.78 |
| IR(↑): 1.172 | IR(↑): -0.2159 | IR(↑): 0.4724 | IR(↑): 0.3245 | IR(↑): 0.2325 | IR(↑): 0.3761 | IR(↑): 1.217 | IR(↑): 1.226 |

[DCI] Prompt: *It is a close up shot of the outdoor sign of an Italian restaurant, on a brick building. The building is made up of long horizontal rectangle grey bricks.*

[DCI] Prompt: *Red flowers hang from a tree branch. There are other trees branches throughout the image.*

*Figure 6.* Qualitative visual results on Qwen-Image with different sparsity strategies over sDCI. FID and IR are computed on the full datasets. Our activation sparsity consistently preserves higher visual quality than competing methods.

*Table 2.* Ablation study of our proposed methods on different DiT models. Here, SA denotes Sparse Activation, NC denotes Norm Compensation, LoRA indicates LoRA adaptation, and SL refers to layer skipping in single-stream paths.

| Model | Method | MJHQ | | | | sDCI | | | |
|---|---|---|---|---|---|---|---|---|---|
| | | FID(↓) | IR(↑) | C.SCR(↑) | C.IQA(↑) | FID(↓) | IR(↑) | C.SCR(↑) | C.IQA(↑) |
| Qwen-Image | Full | 21.98 | 1.219 | 27.12 | 0.9317 | 31.15 | 1.172 | 26.49 | 0.9492 |
| | SA-Native | 35.85 | 0.5994 | 26.28 | 0.7473 | 48.59 | 0.4724 | 26.13 | 0.7550 |
| | SA-NC | 25.28 | 0.9393 | 26.76 | 0.8349 | 37.56 | 0.8399 | 26.34 | 0.8321 |
| | SA-NC-LoRA | **21.25** | **1.304** | **27.33** | **0.9263** | **25.78** | **1.226** | **26.61** | **0.9366** |
| Qwen-Image (2512) | Full | 20.45 | 1.290 | 26.45 | 0.9569 | 24.42 | 1.130 | 25.91 | 0.9435 |
| | SA-Native | 39.13 | 0.6575 | 25.60 | 0.7379 | 51.41 | 0.3205 | **26.48** | 0.6593 |
| | SA-NC | 26.46 | 1.064 | 26.08 | 0.8891 | 33.29 | 0.8747 | 26.32 | 0.8251 |
| | SA-NC-LoRA | **20.84** | **1.302** | **26.54** | **0.9484** | **24.10** | **1.132** | 26.07 | **0.9285** |
| Flux.1-dev | Full | 22.05 | 1.010 | 26.51 | 0.9401 | 25.96 | 1.074 | 26.01 | 0.9487 |
| | SA-Native | 49.87 | 0.4520 | 24.83 | 0.7881 | 52.88 | 0.590 | 25.74 | 0.7471 |
| | SA-NC | 38.67 | 0.7823 | 25.56 | 0.8565 | 39.67 | 0.8704 | 25.86 | 0.8340 |
| | SA-NC-LoRA | 22.61 | 0.9783 | 26.30 | 0.9341 | 26.35 | 1.050 | 25.90 | 0.9415 |
| | SA-NC-LoRA-SL | **21.17** | **1.011** | **26.42** | **0.9381** | **24.41** | **1.091** | **25.90** | **0.9445** |
| Z-image | Full | 25.70 | 0.9928 | 25.57 | 0.9307 | 25.40 | 0.9974 | 25.91 | 0.9467 |
| | SA-Native | 42.65 | 0.5982 | 25.42 | 0.7917 | 36.18 | 0.7306 | **26.72** | 0.7508 |
| | SA-NC | 31.65 | 0.8054 | **25.61** | 0.8525 | 28.48 | 0.8818 | 26.45 | 0.8504 |
| | SA-NC-LoRA | 27.39 | 0.9292 | 25.48 | 0.9016 | 25.98 | 0.9788 | 26.08 | 0.9137 |
| | SA-NC-LoRA-SL | **26.17** | **0.9673** | 25.39 | **0.9250** | **24.93** | **0.9803** | 26.22 | **0.9396** |

tual quality, and semantic alignment. The full protocol is given in Appendix F.4.

**Implementation Details**. All training experiments are conducted on NVIDIA H20 GPUs. The environment uses NVIDIA Driver 580.82.07 and CUDA 13.0. All implementation details are deferred to Appendix F.5, covering online sparse kernels, LoRA fine-tuning, and orthogonality with FP8 quantization and distillation.

### 5.2. Accuracy

**Comparison with SOTA Weight Sparsification**. To compare activation and weight sparsity in DiT inference, we evaluate both on Qwen-Image over MJHQ and sDCI. As shown in Table 1 and Figure 6, weight sparsification causes

severe quality degradation: the naive Sparse Weight baseline performs worst, and Wanda, RIA, and BaWA remain far below the dense model. Slim adapts a similar Lora strategy as ours, it uses a rank of $R = 0.1 * d$, resulting in inference overhead much higher than ours. Using only $R = 64$, our method achieves consistently better results, even surpassing the FP16 model on both benchmarks. Qualitative results are shown in Figure 6. Only our results can generate results nearly indistinguishable from the original. These findings indicate that RT-Lynx preserves critical features that weight pruning irreversibly discards (see Appendix G.1 for more results).

**Ablation**. We conduct ablation studies to evaluate the impact of each proposed component across various DiT architectures (Table 2). While activation sparsity (SA-Native) in-

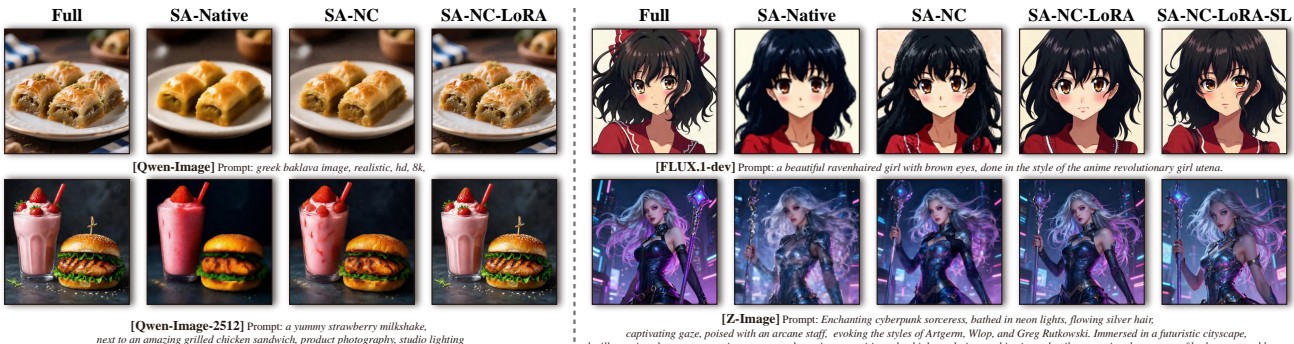

*Figure 7.* Visual ablation results of four categories of models on the MJHQ dataset; more qualitative results are provided in Appendix G.2. Here, SA denotes Sparse Activation, NC denotes Norm Compensation, LoRA indicates the use of LoRA adaptation, and SL refers to layer skipping in single-stream models.

*Table 3.* Compatibility of RT-Lynx with distillation (step reduction from 50 to 8), quantization (W8A8), cache (TeaCache (Liu et al., 2025a) with $l = 0.6$), and sparse attention (SpargeAttn (Zhang et al., 2025d) with $k = 0.5$) in terms of quantitative image quality.

| Model | Method | MJHQ | | | | sDCI | | | |
|---|---|---|---|---|---|---|---|---|---|
| | | FID(↓) | IR(↑) | C.SCR(↑) | C.IQA(↑) | FID(↓) | IR(↑) | C.SCR(↑) | C.IQA(↑) |
| Distillation (Z -Image) | Baseline(step-50) | 22.31 | 0.8790 | 26.25 | 0.9046 | 18.60 | 0.9545 | 26.23 | 0.9196 |
| | *RT-Lynx* | *22.68* | *0.9301* | *26.25* | *0.9058* | *18.80* | *1.006* | *26.25* | *0.9167* |
| | Turbo(step-8) | 25.70 | 0.9928 | 25.57 | 0.9307 | 25.40 | 0.9974 | 25.91 | 0.9467 |
| | **RT-Lynx+Turbo** | **26.17** | **0.9673** | **25.39** | **0.9250** | **24.93** | **0.9803** | **26.22** | **0.9396** |
| Quant (Qwen -Image) | Baseline | 21.98 | 1.219 | 27.12 | 0.9317 | 31.15 | 1.172 | 26.49 | 0.9492 |
| | *RT-Lynx* | *21.25* | *1.304* | *27.33* | *0.9263* | *25.78* | *1.226* | *26.61* | *0.9366* |
| | W8A8 | 21.78 | 1.220 | 27.10 | 0.9298 | 30.59 | 1.173 | 26.48 | 0.9474 |
| | **RT-Lynx+W8A8** | **21.43** | **1.299** | **27.33** | **0.9250** | **25.92** | **1.227** | **26.65** | **0.9341** |
| Cache (FLUX.1 -dev) | Baseline | 22.05 | 1.010 | 26.51 | 0.9401 | 25.96 | 1.074 | 26.01 | 0.9487 |
| | *RT-Lynx* | *21.17* | *1.011* | *26.42* | *0.9381* | *24.41* | *1.091* | *25.90* | *0.9445* |
| | Teacache(l=0.6) | 20.67 | 0.937 | 25.83 | 0.9373 | 25.76 | 1.024 | 25.46 | 0.9487 |
| | **RT-Lynx+Teacache** | **20.57** | **0.941** | **25.73** | **0.9414** | **24.98** | **1.026** | **25.32** | **0.9443** |
| Attention (Qwen -Image -2512) | Baseline | 20.45 | 1.290 | 26.45 | 0.9569 | 24.42 | 1.130 | 25.91 | 0.9435 |
| | *RT-Lynx* | *20.84* | *1.302* | *26.54* | *0.9484* | *24.10* | *1.132* | *26.07* | *0.9285* |
| | SpargeAttn(k=0.5) | 21.17 | 1.192 | 26.18 | 0.9557 | 25.46 | 1.067 | 26.16 | 0.9390 |
| | **RT-Lynx+SpargeAttn** | **21.39** | **1.212** | **26.37** | **0.9420** | **25.22** | **1.075** | **26.42** | **0.9267** |

herently outperforms weight sparsity (Table 1), a substantial performance gap remains. Progressively integrating Norm Compensation (NC) and LoRA (SA-NC-LoRA) effectively closes this gap, restoring FID to 21.25. For single-stream models, Layer Skipping (SL) further optimizes generation. Qualitative results (Figure 7) confirm that our combined strategy successfully mitigates artifacts across all model scales.

**Compatibility with Other Acceleration Methods.** We evaluate whether RT-Lynx can be combined with representative acceleration pipelines without degrading generation quality, including step distillation, W8A8 quantization, feature caching, and sparse attention. As shown in Table 3, adding RT-Lynx on top of these methods preserves comparable quantitative quality across MJHQ and sDCI. In particular, on the 8-step distilled Z-Image model (Table 14), traditional weight pruning leads to catastrophic collapse (FID 360.2), whereas RT-Lynx maintains near-lossless quality compared with the distilled baseline (FID 26.17 vs. 25.70).

Similar quality-preserving behavior is observed when combining RT-Lynx with W8A8 quantization on Qwen-Image, TeaCache on FLUX.1-dev, and SpargeAttn on Qwen-Image-2512. These results show that RT-Lynx does not interfere with step-level, numerical, caching-based, or attention-level optimizations, demonstrating its robustness as a plug-and-play component for diffusion inference.

### 5.3. SpeedUp

**Internal Details**. Benchmarking (Table 4) shows that our framework overcomes the "online sparsity challenge"—the overhead of dynamic sparse generation—by limiting it to less than 10%, yielding up to 1.88× speedup, even as existing backends often falter. These kernel-level gains translate into pervasive layer-level improvements; as illustrated in the Qwen-Image profiling (Figure 8(a)), the approach reduces

---

[*]Due to limited PyTorch support for 2:4 pruning, pruning time is excluded; actual online latency is higher than reported.

*Table 4.* Performance of Dense and Sparse GEMM Backends on H20 GPUs. Other architectural results are provided in Appendix H. Values show runtime (ms); numbers in parentheses denote speedup over GEMM. PyTorch-SpMM measures the online N:M sparse GEMM time using PyTorch. Sparse-Cost reports the proportion of sparse overhead in the total online sparse execution. For the sparse cost of other methods, refer to Table 5. Gray rows indicate matrix sizes used in Qwen-Image.

| $M{=}N$ | $K$ | Pytorch-GEMM | Pytorch-SpMM | CUTLASS | cuSparseLt | RT-Lynx-Kernel | Sparse-Cost(%) |
|---|---|---|---|---|---|---|---|
| 2048 | 3072 | 0.199 | 0.221 (0.90×) | 0.216 (0.92×) | 0.282 (0.71×) | 0.135 (**1.47×**) | **8.33%** |
| 4096 | 3072 | 0.781 | 0.715 (1.09×) | 0.616 (1.26×) | 0.709 (1.10×) | 0.465 (**1.68×**) | **4.60%** |
| 8192 | 3072 | 2.994 | 2.087 (1.43×) | 2.044 (1.46×) | 2.066 (1.50×) | 1.802 (**1.66×**) | **2.28%** |
| 2048 | 12288 | 0.781 | 1.259 (0.62×) | 0.801 (0.97×) | 1.137 (0.69×) | 0.454 (**1.72×**) | **9.39%** |
| 4096 | 12288 | 3.099 | 2.648 (1.17×) | 2.287 (1.35×) | 2.669 (1.16×) | 1.652 (**1.88×**) | **4.83%** |
| 8192 | 12288 | 11.95 | 8.217 (1.45×) | 7.497 (1.59×) | 8.202 (1.45×) | 6.754 (**1.77×**) | **2.37%** |

*Table 5.* Proportion of online activation sparsification overhead in total sparse execution for a typical DiT inference pipeline across methods on H20 GPUs. Gray rows indicate matrix sizes used in Qwen-Image.

| $M{=}N$ | $K$ | PyTorch[*] | CUTLASS | cuSparseLt |
|---|---|---|---|---|
| 2048 | 3072 | 35.70% | 39.44% | 58.41% |
| 4096 | 3072 | 35.08% | 26.98% | 44.61% |
| 8192 | 3072 | 24.63% | 16.02% | 27.80% |
| 2048 | 12288 | 48.65% | 40.98% | 59.46% |
| 4096 | 12288 | 39.29% | 28.10% | 43.37% |
| 8192 | 12288 | 23.14% | 17.01% | 26.99% |

*Table 6.* Orthogonality of RT-Lynx with distillation (step reduction from 50 to 8), quantization (W8A8), cache (TeaCache (Liu et al., 2025a) with $l = 0.6$), and sparse attention (SpargeAttn (Zhang et al., 2025d) with $k = 0.5$) in terms of end-to-end latency.

| Model | Method | End2End(s) |
|---|---|---|
| Distillation (Z-Image) | Baseline(step-50) | 49.44 |
| | *RT-Lynx* | *40.76 (1.21×)* |
| | Turbo(step-8) | 4.99 (9.91×) |
| | **RT-Lynx+Turbo** | **4.17 (11.86×)** |
| Quant (Qwen-Image) | Baseline | 60.66 |
| | *RT-Lynx* | *50.60 (1.20×)* |
| | W8A8 | 54.45 (1.11×) |
| | **RT-Lynx+W8A8** | **46.04 (1.32×)** |
| Cache (FLUX.1-dev) | Baseline | 77.99 |
| | *RT-Lynx* | *63.00 (1.24×)* |
| | TeaCache(l=0.6) | 29.56 (2.64×) |
| | **RT-Lynx+TeaCache** | **24.89 (3.13×)** |
| Attention (Qwen-Image -2512) | Baseline | 60.42 |
| | *RT-Lynx* | *49.35 (1.22×)* |
| | SpargeAttn(k=0.5) | 54.49 (1.11×) |
| | **RT-Lynx+SpargeAttn** | **44.35 (1.36×)** |

latency across all linear layers, with MLP Down projection dropping from 2.22ms to 1.39ms and aggregate linear layer time decreasing from 6.74ms to 4.83ms. Ultimately, these results demonstrate that our framework effectively bridges the gap between theoretical activation sparsity and practical system-level throughput across the entire Transformer architecture. Detailed layer-wise analyses for FLUX.1-dev and Z-Image are provided in Appendix H.

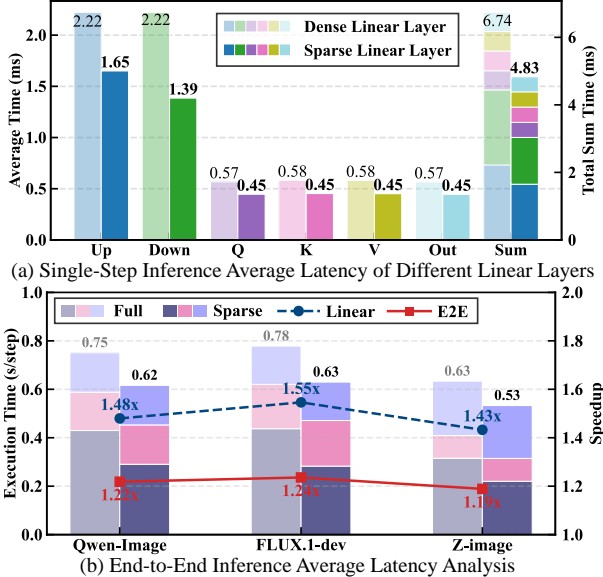

(a) Single-Step Inference Average Latency of Different Linear Layers

(b) End-to-End Inference Average Latency Analysis

*Figure 8.* Latency Analysis. (a) Average per-layer single-step latency of different linear layer types in Qwen-Image, under dense and sparse settings. Sum is the average total latency of one Transformer layer per step. (b) Per-step latency of full image generation across models, with time breakdowns for Linear(bottom), Attention(middle), and other operations(top).

**End-to-End**. Benchmarking on Qwen-Image, FLUX.1-dev, and Z-image (Figure 8(b)) demonstrates that kernel-level efficiencies consistently translate into tangible gains in end-to-end image generation. Our method achieves stable E2E speedups of about 1.2×, driven primarily by the acceleration of computation-dominant Linear layers, which attain speedups between 1.43× and 1.55×. By directly targeting these critical bottlenecks in DiT architectures, the framework substantially reduces per-step latency in raw image synthesis—for example, reducing the generation time of Qwen-Image from 0.75s to 0.62s. This behavior indicates that operator-level optimizations are effectively propagated to the system level, yielding concrete throughput improvements for large-scale generative models during real-time image generation.

**Orthogonality Acceleration with Other Methods.** RT-Lynx is designed to accelerate GEMM-dominated linear layers in DiTs and is fully complementary to other inference optimizations, including step distillation, quantization, caching, and attention acceleration. As shown in Table 6, RT-Lynx consistently composes with these methods to deliver additional end-to-end speedups. For instance, combin-

ing RT-Lynx with 8-step distillation (Turbo) on Z-Image achieves an 11.86× speedup over the original 50-step baseline, compared to 9.91× for Turbo alone. Similar gains are observed with W8A8 quantization on Qwen-Image (1.32× vs. 1.11×), TeaCache on FLUX.1-dev (3.13× vs. 2.64×), and SpargeAttn on Qwen-Image-2512 (1.36× vs. 1.11×). These results demonstrate that RT-Lynx can be seamlessly integrated into diverse acceleration pipelines, consistently improving end-to-end efficiency. Such composability makes RT-Lynx a practical and effective plug-and-play module for further accelerating diffusion model inference.

## 6. Related Work

**Model Sparsity for LLMs**. Weight structured sparsity enforces explicit structural constraints on parameters so that sparse patterns can be mapped onto hardware-friendly execution units, reducing effective computation and improving throughput (Frantar & Alistarh, 2023; Sun et al., 2023; Zhang et al., 2024a; Liu et al., 2025g; Mozaffari et al., 2025). Training-free methods such as SparseGPT (Frantar & Alistarh, 2023), Wanda (Sun et al., 2023), RIA (Zhang et al., 2024a), and BaWA (Mozaffari et al., 2025) adopt one-shot calibration based on statistics to determine the sparse masks. Subsequent work refines these heuristics with more principled criteria, including sensitivity analysis (Zhang et al., 2024b), gradient-based estimation (Das et al., 2024), entropy modeling (Li et al., 2024c), and low-rank–aware scoring (Zhang & Papyan, 2024), improving pruning stability and effectiveness. Learning-driven approaches further parameterize sparsity via optimizable masks or auxiliary modules, as in MaskLLM (Fang et al., 2024) and ProxSparse (Liu et al., 2025c), while training-coupled methods such as SLoPe (Mozaffari et al., 2024) and AST (Huang et al., 2025) enforce structure throughout optimization to obtain stable sparsity.

**Activation Sparsity**. In contrast to weight sparsity, activation sparsity remains under-explored. Previous works primarily focus on decoding activation sparsity, which prunes sparse channels in the corresponding weight (Mirzadeh et al., 2023; Song et al., 2024; Zhang et al., 2024c; Lee et al., 2024; Song et al., 2025; Liu et al., 2024; Wang et al., 2024; Liu et al., 2025f). However, these methods are limited to the case where the total token count is below 4, making them inapplicable to DiTs where token counts typically exceed 1,000. The most relevant works are An et al. (2025) and Haziza et al. (2025). In An et al. (2025), authors propose to use an 8:16 sparse pattern that has not been supported on current GPUs, while Haziza et al. (2025) applies activation sparsity to LLM pretraining but limits sparsification to FFN layers. Both works focus exclusively on LLMs and do not address the unique challenges posed by diffusion models.

**Diffusion Acceleration**. Numerous approaches have been proposed for accelerating diffusion models (Yuan et al.,

2024; Zhang et al., 2025b; Xi et al., 2025; Ma et al., 2023; Selvaraju et al., 2024; Chen et al., 2024b; Ma et al., 2025; Liu et al., 2025e), including quantization, distillation, sparse attention, and feature caching. SVDQuant (Li et al., 2025) applies 4-bit quantization on DiTs, while Yang et al. (2024) further pushes the quantization level to 1.58bits. For step distillation, representative works include distributional matching distillation (Yin et al., 2024) and consistency distillation (Zheng et al., 2025; Lu & Song, 2025). To reduce the computation cost of full attention, Chen et al. (2025); Fang et al. (2026) propose replacing the original attention mechanism with linear attention, while Zhang et al. (2025c;e) explore using sparse attention to approximate the original output. Additionally, computational costs can be reduced by caching features across diffusion steps, as demonstrated by TeaCache (Liu et al., 2025b) and TaylorSeer (Liu et al., 2025d). Unlike these approaches, our work focuses on exploiting activation sparsity in DiTs, offering an orthogonal and complementary acceleration strategy.

## 7. Conclusion

In this work, we leverage semi-structured sparsity to accelerate inference in DiT models by shifting the sparsification paradigm from weights to activations. Empirical analysis reveals that DiT activations are intrinsically sparse due to superposition and exhibit substantially greater robustness under N:M sparsity than weights. Building on this insight, we propose RT-Lynx, which applies N:M sparsification to intermediate activations, incorporates norm-based error compensation, a lightweight LoRA branch, and layer skipping to reduce the sparsification error. At the system level, we develop highly optimized CUDA kernels that efficiently generate and exploit online activation sparsity during inference. Extensive experiments across multiple diffusion models demonstrate that RT-Lynx preserves generation quality while substantially accelerating inference, achieving up to a 1.55× average speedup in linear layers. In this context, to our knowledge, this is the first work to achieve lossless N:M sparsification for DiT models with substantial speedups.

## Acknowledgment

We sincerely thank the anonymous reviewers for their insightful suggestions. The work is supported by the Natural Key Research and Development Program of China (2023YFB3002902) and the Fundamental Research Funds for the Central Universities(JKF20240567). Hanlin Tang is the project leader. Chenhao Xie is the corresponding author (xiechenhao@buaa.edu.cn)

## Impact Statement

This paper presents work whose goal is to improve the inference efficiency of diffusion models. There are many potential societal consequences of more efficient generative AI systems, none of which we feel must be specifically highlighted here.

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

## A. Sparse Tensor Core in NVIDIA

NVIDIA's Sparse Tensor Core (SpTC) is a dedicated execution path built on top of Tensor Cores to support semi-structured sparsity. Its objective is to increase computational density by reducing the number of effective multiplications, while preserving regular dataflow and the existing hardware pipeline. Unlike general sparse computation, SpTC does not support arbitrary sparsity patterns; instead, it is strictly constrained to fixed, hardware-predictable, and encodable structures.

*Table 7.* Supported Sparse Formats and Precisions in NVIDIA Sparse Tensor Cores

| Format | Precision |
|--------|-----------|
| 1:2 | TF32 |
| 2:4 | FP16, BF16, FP8, INT8 |
| 4:8 | INT4, FP4 |

The semi-structured sparsity formats currently supported by NVIDIA are summarized in Table ref tab:sptc$_f ormats, among which the 2: 4 sparsity pattern is the most representative and has been practically deployed.$

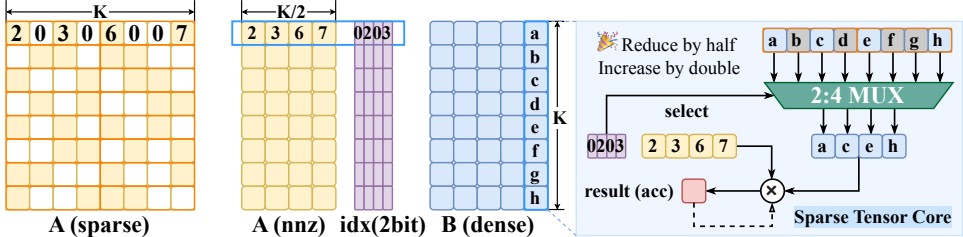

*Figure 9.* SpTC Calculation Diagram

As illustrated in the figure9, during the execution of 2:4 sparsity on SpTC, matrix $A$ retains only two nonzero elements out of every four along the $K$ dimension, which are stored as $A_{\text{nnz}}$ together with a corresponding 2-bit index. During computation, the index is used to select the matching two elements from the four candidate inputs of the dense matrix $B$, which are then multiplied by the nonzero weights and accumulated into the output. This mechanism preserves regular memory access and computation patterns while halving the number of multiplications, thereby enabling, in the ideal case, up to a $2\times$ throughput improvement over Dense Tensor Cores and providing a foundation for direct application-level acceleration.

## B. Experimental Details for Comparing Weight Sparsity and Activation Sparsity

*Table 8.* Tensor Shapes of Weights and Activations in the Image and Text Modules of Qwen-Image (where $X$ denotes activations, $W$ denotes weights, $N_{\text{txt}}$ is the text length, and $N_{\text{img}}$ is the image length). Text in brackets([abbr.]) indicates the abbreviation used in this paper for this type of layer in the model.

| Image Module | Image X shape | Image Weight shape | Text Module | Text X shape | Text Weight shape |
|---|---|---|---|---|---|
| img_mod.1 | (3072) | (3072, 18432) | txt_mod.1 | (3072) | (3072, 18432) |
| attn.to_q [Q] | $(N_{\text{img}}, 3072)$ | (3072, 3072) | attn.add_q_proj | $(N_{\text{txt}}, 3072)$ | (3072, 3072) |
| attn.to_k [K] | $(N_{\text{img}}, 3072)$ | (3072, 3072) | attn.add_k_proj | $(N_{\text{txt}}, 3072)$ | (3072, 3072) |
| attn.to_v [V] | $(N_{\text{img}}, 3072)$ | (3072, 3072) | attn.add_v_proj | $(N_{\text{txt}}, 3072)$ | (3072, 3072) |
| attn.to_out.0 [Out] | $(N_{\text{img}}, 3072)$ | (3072, 3072) | attn.to_add_out | $(N_{\text{txt}}, 3072)$ | (3072, 3072) |
| img_mlp.net.0.proj [Up] | $(N_{\text{img}}, 3072)$ | (3072, 12288) | txt_mlp.net.0.proj | $(N_{\text{txt}}, 3072)$ | (3072, 12288) |
| img_mlp.net.2 [Down] | $(N_{\text{img}}, 12288)$ | (12288, 3072) | txt_mlp.net.2 | $(N_{\text{txt}}, 12288)$ | (12288, 3072) |

Building on the preceding observation that activation sparsity affords greater flexibility than weight sparsity, we conduct a set of comparative experiments on the Qwen-Image model (architectural details are provided in the Appendix F.1). Image quality is evaluated using ImageReward, with a detailed description of the metric deferred to the Appendix F.4. The types of Linear layers contained in the Transformer blocks of Qwen-Image are summarized in Table 8. Following the modules specified in Algorithm 2, we replace all eligible linear layers in the DiT model, enabling a controlled evaluation of weight sparsity and dynamic activation sparsity, respectively, during inference.

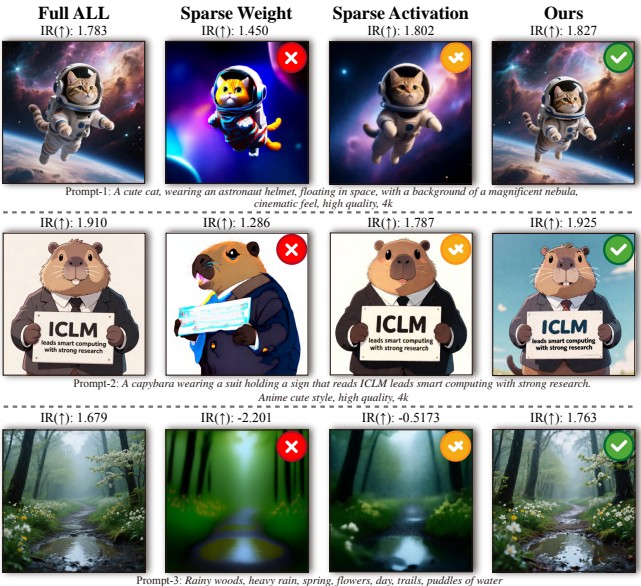

*Figure 10.* Comparison of sparsification strategies on Qwen-Image. The figure illustrates the visual impact of different methods on generation quality. Both weight sparsity and naive activation sparsity apply Top-$K$ selection to enforce 2:4 semi-structured sparsity. IR($\uparrow$) reports the Image Reward score (Xu et al., 2023) for each image.

---

**Algorithm 2** 2:4 Sparsity for Linear Layers

---

**Input:** DiT network $\mathcal{M}$, mode $m \in \{$full, weight_sparse, activation_sparse$\}$
**Output:** Modified network $\mathcal{M}$ with 2:4 sparse linear layers
**Forward of each replaced layer** $\tilde{C}$
**if** $m =$ weight_sparse **then**
    Reshape $W$ into blocks of size 4, In each block keep two entries with the largest magnitude, zero the others to get $\tilde{W}$
    $y \leftarrow x\tilde{W}^{\top} + b$
**else if** $m =$ activation_sparse **then**
    Reshape $x$ into blocks of size 4, In each block keep two entries with the largest magnitude, zero the others to get $\tilde{x}$
    $y \leftarrow \tilde{x}W^{\top} + b$
**end if**
**Procedure: Replace Linear layers in** $\mathcal{M}$
**for** each child module $C$ in $\mathcal{M}$ **do**
    **if** $C$ is a Linear layer and satisfies constraints **then**
        Create a new layer $\tilde{C}$ with copied parameters $(W_C, b_C)$
        Set $\tilde{C}.mode \leftarrow m$
        Replace $C$ by $\tilde{C}$ in $\mathcal{M}$
    **else**
        Recursively apply this procedure to submodules of $C$
    **end if**
**end for**

---

In the experimental setup, mod-related modules are excluded since they perform dynamic feature modulation and do not involve matrix operations. We retain only the text and image data flows. As ($N_{\text{txt}}$) is typically small (no more than 512 in standard DiT models), our analysis focuses on sparsity in the image data flow. We therefore consider two configurations. The first applies sparsity only to the image branch (Figure 1), which is used throughout the paper. The second applies sparsity to both image and text branches (Figure 10). In this setting, activation sparsity shows a clearer advantage over weight sparsity, further indicating that **activation sparsity has a smaller impact on output quality**. However, because the text branch is small, sparsifying it brings negligible performance gains, and we do not adopt this configuration in our paper's experiments.

## C. Additional Results of Rethinking Sparsity

In the supplementary experiments, we observe results consistent with those of the main study. Across different Linear layer types, weight values generally exhibit a more uniform distribution, whereas activations remain highly concentrated around zero, as shown in Figure11 (a) and Figure11 (b). Although this contrast is weaker in the MLP Up layers, it persists in most other linear operators.

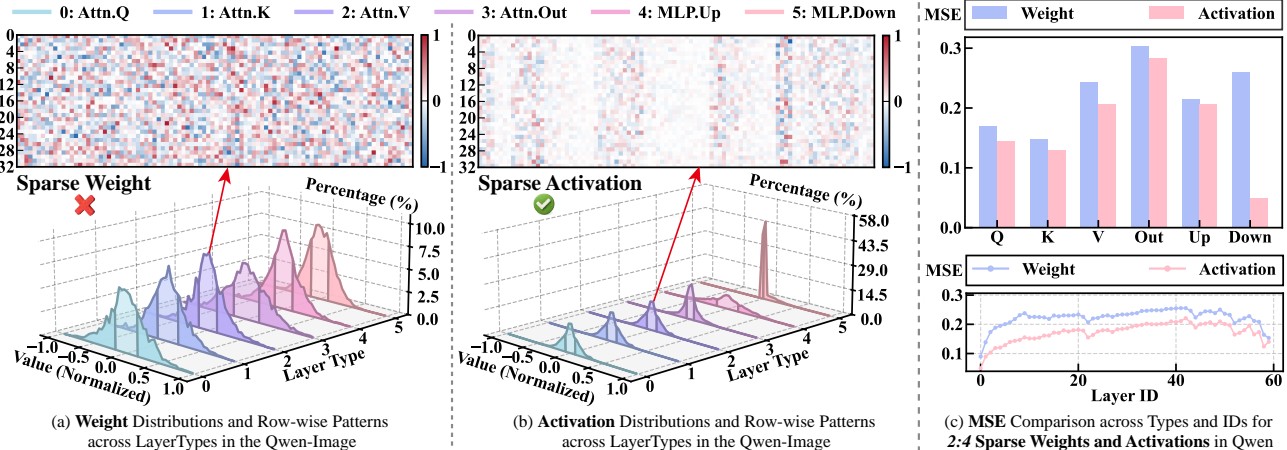

(a) **Weight** Distributions and Row-wise Patterns across LayerTypes in the Qwen-Image

(b) **Activation** Distributions and Row-wise Patterns across LayerTypes in the Qwen-Image

(c) **MSE** Comparison across Types and IDs for *2:4 Sparse Weights and Activations* in Qwen

*Figure 11.* Distributions and sparsity-induced errors across Linear layers in DiT. (a)–(b) compare weights and activations at the 6th layer of Qwen-Image: the lower parts show normalized value distributions for six Linear layer types (with [-1,1] divided into 50 bins and lighter regions indicating values below 50% of the mass), while the upper parts visualize normalized heatmaps of the same row from the Attention.V projection, where lighter colors denote values closer to zero. (c) reports the average relative errors induced by Top-$K$ 2:4 sparsification across Linear layer types (top) and layer depths (bottom).

Combined with depth-wise analysis, these findings indicate that, under the same 2:4 structured sparsity, weight sparsification—regardless of layer type or depth—removes more parameters that still contribute to the output, whereas activation sparsification primarily eliminates low-magnitude components and thus has a much smaller direct impact. This distinction is also evident at the row level: weight matrices tend to be row-dense with similar magnitudes, whereas activations exhibit inherently sparse row structures that better align with 2:4 constraints, leading to lower instantaneous representation error.

The error statistics in Figure11 (c) further corroborate this behavior, showing consistently higher mean squared errors for weight sparsification across both layer types and depths, while activation sparsification yields uniformly lower and more stable errors, reinforcing the core conclusions of this study.

## D. Training Details and Rank Selection

### D.1. Training Details

All experiments are conducted within the **diffsynth-studio** framework provided by ModelScope (ModelScope Community, 2025).

**Process**. During training, we adhere to the parameter-efficient fine-tuning paradigm and introduce sparse LoRA adaptation branches only at the critical linear projection layers of the diffusion backbone, while keeping all other parameters frozen. Each LoRA branch is instantiated with a fixed rank and initialized from a uniform distribution over $[-10^{-5}, 10^{-5}]$, ensuring that the injected parameters remain vanishingly small at the outset. This design renders the initial perturbation to the base model negligible, thereby guaranteeing that the student model is functionally identical to the base model at initialization. Sparsity is imposed via structured constraints on LoRA updates, forcing the parameters to lie in a low-rank subspace while adhering to a predefined sparsity pattern. This design directly aligns training with structured acceleration at inference. As gradients act solely on LoRA parameters, optimization is confined to a compact subspace, reducing degrees of freedom and alleviating overfitting.

**Datasets**. Training data are constructed by leveraging a large language model to automatically generate diverse user prompts (Peinl, 2025). Compared to manually designed templates, this approach achieves significantly broader semantic coverage and richer linguistic variability, more faithfully approximating the distribution of real-world user inputs. As a result, the

trained model exhibits improved generalization in downstream deployment scenarios.

**Loss**. Supervision is provided via a distillation objective (Lipman et al., 2023) within the diffusion framework. We instantiate a frozen teacher model whose parameters are identical to those of the student at initialization and remain fixed throughout training. For each training sample, a diffusion timestep $t$ is uniformly sampled, and Gaussian noise $\epsilon \sim \mathcal{N}(0, I)$ is injected into the latent variable $z_0$, yielding the noisy representation

$$z_t = \alpha_t z_0 + \sigma_t \epsilon \tag{6}$$

where $\alpha_t$ and $\sigma_t$ are determined by the diffusion scheduler. We denote $\mathbf{X} = z_t$. The student computes its intermediate representation via the proposed sparse–residual decomposition,

$$\mathbf{Y} = \mathbf{Y_s} + \mathbf{Y_r} = S(\mathbf{X}) \cdot \mathbf{W}^\top + \mathbf{X} \cdot (\mathbf{L_A L_B})^\top \tag{7}$$

where $S(\cdot)$ denotes a structured 2:4 sparsification operator applied to the activation, $\mathbf{W}$ is the frozen base weight, and $\mathbf{L_A L_B}$ is the trainable low-rank adapter. The term $\mathbf{Y_s}$ captures the response of the original model under sparse activation, while $\mathbf{Y_r}$ provides a dense low-rank residual that compensates for the error induced by sparsity. Only $\mathbf{L_A}$ and $\mathbf{L_B}$ are updated during training. The frozen teacher always operates on dense activations and the full weight,

$$\mathbf{Y}_{\text{teacher}} = \mathbf{X} \cdot \mathbf{W}^\top \tag{8}$$

and predicts the noise field $\hat{\epsilon}_{\text{teacher}}(z_t, t)$ under a no-gradient regime. The student prediction $\hat{\epsilon}_{\text{student}}(z_t, t)$ is obtained from $\mathbf{Y}$ in Eq. (7). Their discrepancy is measured by a weighted mean squared error with a timestep-dependent weighting function $w(t)$:

$$\mathcal{L} = w(t) \left\| \hat{\epsilon}_{\text{student}}(z_t, t) - \hat{\epsilon}_{\text{teacher}}(z_t, t) \right\|_2^2 \tag{9}$$

This objective encourages the student to reproduce the teacher's noise prediction behavior while any deviation introduced by the sparse activation $S(\mathbf{X})$ in the base path is compensated exclusively through the low-rank branch $\mathbf{L_A L_B}$. As a result, all learning capacity is confined to the LoRA subspace, and the base weight $\mathbf{W}$ together with its induced diffusion dynamics remains unchanged. Throughout training, we periodically perform inference under a fixed scheduling policy to generate sample images, enabling continuous monitoring of the impact of sparse LoRA on generation quality and convergence behavior.

### D.2. Rank Selection

A critical hyperparameter in low-rank adaptation is the rank $R$, which dictates the dimensionality of the update matrices. While a higher rank theoretically provides greater representation capacity, it simultaneously introduces increased computational overhead. Therefore, identifying an optimal trade-off between generative fidelity and adaptation cost is paramount, especially after incorporating activation sparsification.

*Table 9.* Comprehensive ablation study of LoRA ranks on Qwen-Image. We report generative quality (FID), human preference (IR), and semantic alignment (C.SCR, C.IQA) across MJHQ and sDCI datasets. Bold values indicate the best performance within each training step group.

| Step | Rank | MJHQ | | | | sDCI | | | |
|------|------|------|------|------|------|------|------|------|------|
| | | FID($\downarrow$) | IR($\uparrow$) | C.SCR($\uparrow$) | C.IQA($\uparrow$) | FID($\downarrow$) | IR($\uparrow$) | C.SCR($\uparrow$) | C.IQA($\uparrow$) |
| Full | Full | 21.98 | 1.219 | 27.12 | 0.9317 | 31.15 | 1.172 | 26.49 | 0.9492 |
| 200 | 32 | 22.44 | **1.312** | **27.33** | 0.9236 | 26.40 | 1.243 | 26.64 | 0.9468 |
| | 64 | **22.20** | 1.281 | 27.26 | **0.9261** | 26.64 | **1.245** | **26.65** | **0.9483** |
| | 128 | 22.49 | 1.289 | 27.24 | 0.9231 | **25.88** | 1.232 | 26.54 | 0.9401 |
| 2k | 32 | 21.60 | 1.299 | 27.26 | 0.9224 | 28.58 | **1.237** | 26.47 | 0.9342 |
| | 64 | **21.25** | **1.304** | **27.33** | 0.9263 | 25.78 | 1.226 | **26.61** | **0.9366** |
| | 128 | 21.40 | 1.284 | 27.23 | **0.9276** | **25.37** | 1.201 | 26.55 | 0.9361 |
| 20k | 32 | 21.59 | **1.301** | 27.30 | 0.9274 | 25.30 | 1.212 | 26.58 | 0.9364 |
| | 64 | 21.67 | 1.294 | **27.33** | 0.9276 | **24.27** | 1.201 | **26.61** | 0.9306 |
| | 128 | **21.30** | 1.287 | 27.27 | **0.9281** | 25.83 | **1.213** | 26.53 | **0.9395** |

Our multidimensional evaluation in Table 9 and Figure 12 identifies $R = 64$ as the optimal "**sweet spot**" for the proposed framework. Operating at a negligible parameter footprint of 0.7306% and a relative time overhead of 9.808%, $R = 64$ achieves state-of-the-art results on the MJHQ dataset (2k steps), including the lowest FID (21.25) and superior Image-Reward

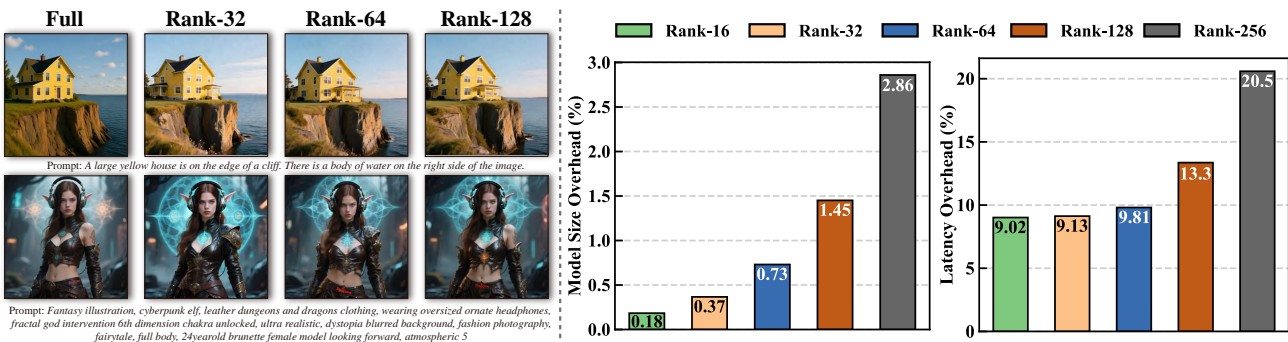

*Figure 12.* Model Size Overhead and Latency Overhead of LoRA with different ranks on Qwen-Image.

(1.304) and CLIP.score (27.33). These results demonstrate that $R = 64$ provides sufficient degrees of freedom to capture complex feature shifts while maintaining high aesthetic quality.In contrast, lower-rank configurations (e.g., $R = 32$) exhibit clear capacity bottlenecks, failing to adequately model the target distribution as evidenced by significantly higher FID scores. Conversely, increasing the rank to $R = 128$ leads to diminishing marginal returns; despite doubling the parameter count and increasing latency to 13.36%, the performance—specifically FID (21.40)—shows signs of saturation or slight degradation. This suggests that excessive rank may introduce redundant noise, potentially compromising the model's generalization. Furthermore, $R = 64$ demonstrates consistent stability across training phases, with CLIP.scores on the sDCI dataset reliably exceeding 26.6. This robust cross-dataset performance, achieved with less than a 1% parameter increment, validates our selection of $R = 64$ as the standard configuration. By precisely targeting task-relevant subspaces, Rank=64 ensures optimal generative fidelity with minimal impact on inference efficiency.

## E. Relative Error Analysis of Activation Sparsity in a Text–Image Hybrid DiT

*Table 10.* Tensor Shapes of Weights and Activations in the Image and Text Streams of FLUX.1-dev (where $X$ denotes activations, $W$ denotes weights, $N_{\text{txt}}$ is the text length, and $N_{\text{img}}$ is the image length). Text in brackets([abbr.]) indicates the abbreviation used in this paper for this type of layer in the model.

| Img Module | Img $X$ shape | Img $W$ shape | Txt Module | Txt $X$ shape | Txt $W$ shape |
|---|---|---|---|---|---|
| norm1_a.linear | $(3072)$ | $(3072, 18432)$ | norm1_b.linear | $(3072)$ | $(3072, 18432)$ |
| attn.a_to_qkv [QKV] | $(N_{\text{img}}, 3072)$ | $(3072, 9216)$ | attn.b_to_qkv | $(N_{\text{txt}}, 3072)$ | $(3072, 9216)$ |
| attn.a_to_out [Out] | $(N_{\text{img}}, 3072)$ | $(3072, 3072)$ | attn.b_to_out | $(N_{\text{txt}}, 3072)$ | $(3072, 3072)$ |
| ff_a.0 [Up] | $(N_{\text{img}}, 3072)$ | $(3072, 12288)$ | ff_b.0 | $(N_{\text{txt}}, 3072)$ | $(3072, 12288)$ |
| ff_a.2 [Down] | $(N_{\text{img}}, 12288)$ | $(12288, 3072)$ | ff_b.2 | $(N_{\text{txt}}, 12288)$ | $(12288, 3072)$ |

| Mixed Module | Mixed $X$ shape | Mixed $W$ shape |
|---|---|---|
| norm.linear | $(3072)$ | $(3072, 9216)$ |
| to_qkv_mlp [QKV Up] | $(N_{\text{img}} + N_{\text{txt}}, 3072)$ | $(3072, 21504)$ |
| proj_out [Out Down] | $(N_{\text{img}} + N_{\text{txt}}, 15360)$ | $(15360, 3072)$ |

To evaluate the sensitivity of different layer types to activation sparsification across models, we conduct a full image generation experiment on two architectures: a hybrid double/single-stream model (FLUX.1-dev) and a single-stream model (Z-Image). We then measure the relative Frobenius error induced by activation sparsification for each layer type, which is defined as

$$\text{RFE} = \frac{\|\mathbf{Y}_{\text{full}} - \mathbf{Y}_{\text{sparse}}\|_F}{\|\mathbf{Y}_{\text{full}}\|_F}. \tag{10}$$

Detailed descriptions of these models are provided in Appendix F.1. The linear layer types contained in the Transformer blocks of each model are summarized in Tables 10 and 11, respectively, and are analyzed in conjunction with Figure 13.

The Figure 13 reports the average relative Frobenius error(RFE) induced by activation sparsity across different layer types under the full text-to-image inference pipeline. Both models exhibit pronounced layer-wise sensitivity, with mixed text–image layers consistently showing higher RFE, confirming that cross-modal shared representations in single-stream paths are particularly vulnerable to sparse perturbations. In FLUX.1-dev, the `Out-Down[mix]` layer attains the highest

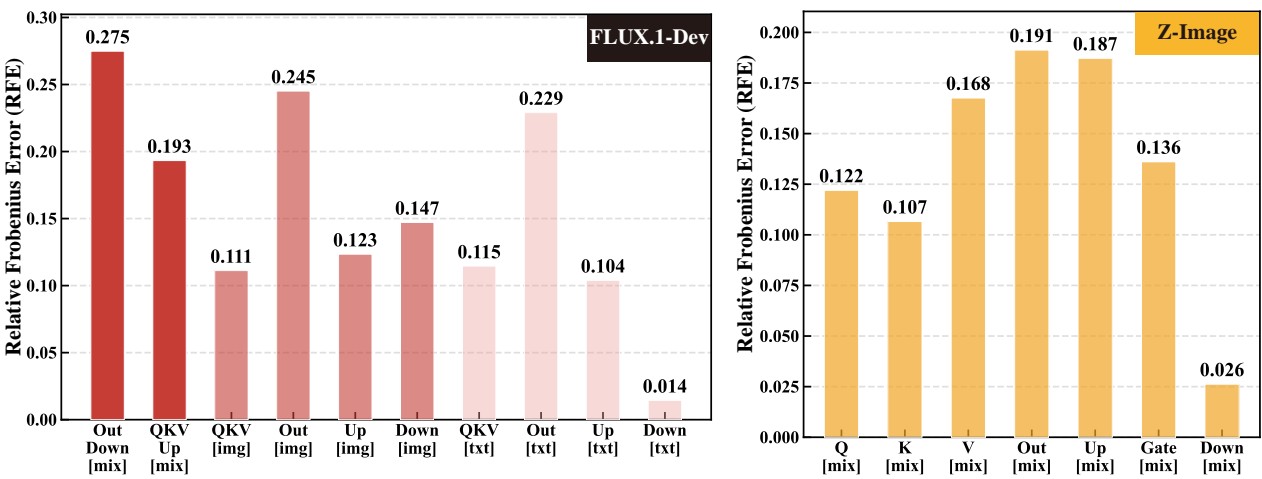

*Figure 13.* Average relative Frobenius error(RFE) of activation sparsity across different layer types in FLUX.1-dev and Z-Image models, with both models executing the full text-to-image generation pipeline.

*Table 11.* Tensor Shapes of Weights and Activations in the Mixed Stream of ZImage (where $X$ denotes activations, $W$ denotes weights, $N_{txt}$ is the text length, and $N_{img}$ is the image length). Text in brackets([abbr.]) indicates the abbreviation used in this paper for this type of layer in the model.

| Mixed Module | Mixed $X$ shape | Mixed $W$ shape |
|---|---|---|
| attention.to_q [Q] | $(N_{img} + N_{txt}, 3840)$ | $(3840, 3840)$ |
| attention.to_k [K] | $(N_{img} + N_{txt}, 3840)$ | $(3840, 3840)$ |
| attention.to_v [V] | $(N_{img} + N_{txt}, 3840)$ | $(3840, 3840)$ |
| attention.to_out.0 [Out] | $(N_{img} + N_{txt}, 3840)$ | $(3840, 3840)$ |
| feed_forward.w1 [Up] | $(N_{img} + N_{txt}, 3840)$ | $(3840, 10240)$ |
| feed_forward.w3 [Gate] | $(N_{img} + N_{txt}, 3840)$ | $(3840, 10240)$ |
| feed_forward.w2 [Down] | $(N_{img} + N_{txt}, 10240)$ | $(10240, 3840)$ |
| adaLN_modulation.0 | $(256)$ | $(256, 15360)$ |

RFE (RFE = 0.275), substantially exceeding `QKV[mix]` and all unimodal layers. This indicates that sparsification at this layer produces the largest relative output deviation and that this deviation propagates along the single-stream path. As this layer jointly performs post-fusion linear projection and downstream MLP mapping, its weight matrix exhibits stronger amplification of sparse perturbations, rendering LoRA ineffective under a limited rank budget. Consequently, selectively skipping the `Out-Down` mixed layer in FLUX.1-dev effectively suppresses the dominant source of cross-modal error. In the purely single-stream Z-Image, all linear layers operate in a fully shared text-image space, resulting in a higher, more concentrated RFE profile. In particular, `Out[mix]` and `Up[mix]` reach RFE = 0.191 and 0.187, respectively, constituting the primary contributors to sparse error. Since Z-Image adopts a turbo-distilled variant and is inherently more sensitive to representational shifts, sparse perturbations more readily disrupt the finely matched distilled distributions. This necessitates a more aggressive structural avoidance strategy: skipping both `Out` and `Up` layers in Z-Image substantially reduces the accumulation of relative deviations in the shared representation space and preserves generation quality.

# F. Experiment Details

## F.1. Benchmark Models

We benchmark our methods using the following four dit models:

- **Qwen-Image**(Wu et al., 2025) is an open-source foundation model for image generation and editing, built on multimodal alignment. It comprises 60 stacked Transformer blocks with roughly 20B parameters. The model adopts a dual-encoder design that aligns high-level semantic representations from Qwen2.5-VL with VAE reconstruction features, and further incorporates an enhanced MMDiT architecture: text and image streams are modeled separately and are cross-fused only at the attention stage. This design enables state-of-the-art performance in complex Chinese/English text rendering, multi-line layout, and consistency in image editing.

- **Qwen-Image-2512**(Wu et al., 2025) is the major annual update of the Qwen-Image series released in December 2025. It ranks first among open-source models in over 10,000 rounds of blind evaluations on AI Arena, substantially improving photorealism and reducing visible "AI artifacts." Compared with the August release, it delivers pronounced gains in natural detail synthesis (e.g., hair and fine textures) and text–image alignment accuracy.

- **FLUX.1**(Labs et al., 2025) is a strong open-source diffusion model based on the Diffusion Transformer (DiT) paradigm, consisting of 19 dual-stream joint-attention blocks and 38 single-stream parallel-attention blocks. The dual-stream modules strengthen interactions between conditioning signals and latent variables, while the single-stream modules provide efficient self-attention modeling and feature transformation. With roughly 12B parameters, FLUX.1 achieves high generation quality and strong text alignment while maintaining scalability for both training and inference. In this work, we evaluate the **FLUX.1-dev** variant.

- **Z-Image**(Team, 2025) is among the most representative efficient open-source image generation models. It adopts a scalable single-stream DiT (S3-DiT) architecture that concatenates text, visual semantics, and image VAE tokens into a unified Transformer stream. Despite having only about 6B parameters, it produces high-quality photorealistic images and robust Chinese/English text rendering, and ranks first among open-source models on the Artificial Analysis leaderboard. We evaluate its distilled variant, **Z-Image-Turbo**, which generates high-quality images with only 8 inference steps.

## F.2. Benchmark Datasets

Consistent with prior work(Xie et al., 2024; Han et al., 2025; Li et al., 2024b; Chen et al., 2024a; Bolya et al., 2025), we evaluate generalization across diverse prompt styles using two complementary benchmarks.

- **MJHQ-30K**(Li et al., 2024a) is a collection of 30K art-oriented prompts sourced from Midjourney, covering ten common artistic categories with a balanced distribution. This dataset primarily reflects highly stylized, aesthetics-driven generation demands. In our experiments, we randomly sample 5,000 prompts to assess performance in artistic image generation.

- **Densely Captioned Images (DCI)**(Urbanek et al., 2024) focuses on real-world imagery with dense semantic descriptions. It contains approximately 8,000 images, each annotated with long, human-written, fine-grained captions averaging over one thousand words. Given practical limits on text length in diffusion models, we use its compressed variant, **sDCI**, where the original descriptions are summarized to 77 tokens using a large language model. From this subset, we likewise randomly sample 5,000 prompts to evaluate generalization in photorealistic image generation.

## F.3. Sparse Weight Method Details

- **Wanda.** Wanda casts pruning in each linear layer with weight matrix $W \in \mathbb{R}^{d_{\text{out}} \times d_{\text{in}}}$ as an element-wise importance estimation based on *weight magnitude* modulated by *input activation strength*. The key intuition is that, for weights of comparable magnitude, those connected to more "active" input channels (i.e., larger activation norms on a calibration set) incur a larger reconstruction penalty if removed and should therefore be retained. This yields a unified scoring rule

$$S_{ij} = |W_{ij}| \cdot A_j,$$

where Wanda instantiates $A_j = \|X_j\|_2$ (with $X$ denoting the stacked calibration activations and $X_j$ the $j$-th input-channel activation vector), giving

$$S_{ij}^{\text{Wanda}} = |W_{ij}| \cdot \|X_j\|_2.$$

Pruning then zeroes entries with the smallest $S_{ij}$ within each output channel to satisfy a target sparsity level or an N:M constraint.

- **RIA.** To mitigate *channel corruption* in post-training sparsification—where certain input/output channels are excessively emptied—RIA introduces a *relative importance* term that depends not only on $|W_{ij}|$ but also on its proportion within both its input and output channels. Using $\ell_1$ row/column normalizations, it defines

$$\mathrm{RI}_{ij} = \frac{|W_{ij}|}{\sum_k |W_{kj}|} + \frac{|W_{ij}|}{\sum_k |W_{ik}|},$$

which suppresses weights that are not salient within their channels despite large absolute magnitudes and reduces the risk of pruning entire channels. RIA then incorporates activations to reflect stable channel significance, yielding the unified form

$$S_{ij} = B_{ij} \cdot A_i^a,$$

where $B_{ij} = \mathrm{RI}_{ij}$ and $A_i = \|X_i\|_2$, hence

$$S_{ij}^{\mathrm{RIA}} = \Big(\frac{|W_{ij}|}{\sum_k |W_{kj}|} + \frac{|W_{ij}|}{\sum_k |W_{ik}|}\Big) \cdot \big(\|X_i\|_2\big)^a,$$

with exponent $a$ controlling the strength of activation modulation.

- **BaWA.** BaWA can be viewed as a principled refinement of magnitude–activation product metrics (e.g., Wanda/RIA) via two systematic adjustments. First, it performs *magnitude normalization* by re-scaling $|W_{ij}|$ with channel-wise weight norms from both input and output sides to alleviate mask imbalance caused by inter-channel scale discrepancies. Second, it applies *outlier regularization* through power-law factors that compress the dynamic range of channel norms, reducing the dominance of activation/weight outliers in pruning decisions. Under the unified scoring template

$$S_{ij} = \big(|W_{ij}| \cdot C_j^{\mathrm{in}} + |W_{ij}| \cdot C_i^{\mathrm{out}}\big) \cdot A_j,$$

BaWA sets $C_j^{\mathrm{in}} = \|W_j\|_2^{-\theta_1}$, $C_i^{\mathrm{out}} = \|W_i\|_2^{-\theta_2}$, and $A_j = \|X_j\|_2^{\theta_3}$, yielding

$$S_{ij}^{\mathrm{BaWA}} = \Big(\frac{|W_{ij}|}{\|W_j\|_2^{\theta_1}} + \frac{|W_{ij}|}{\|W_i\|_2^{\theta_2}}\Big) \cdot \|X_j\|_2^{\theta_3}.$$

The tunable exponents $(\theta_1, \theta_2, \theta_3)$ serve as searchable "balancing knobs" that regulate the attenuation/amplification of input-side weight norms, output-side weight norms, and activation norms, respectively, aiming to produce a more uniform and effective sparse mask under heterogeneous layer/module scales.

- **SLiM.** SLiM does not introduce a new pruning criterion; instead, it applies an off-the-shelf one-shot pruning method to impose unstructured or semi-structured sparsity, producing a sparse weight $W^C$ and an induced sparsification error $E_S$. Its *sparse-weight optimization* then compensates compression error via a low-rank adapter, targeting

$$W \approx W^C + LR.$$

Importantly, unlike standard LoRA that can often be merged into dense weights for efficient inference, SLiM's setting couples an N:M-structured sparse $W^C$ with a *dense* low-rank update $LR$, which generally cannot be fused into the sparse format without breaking the sparsity pattern. Consequently, the low-rank term must be computed online; if the rank is large, it can substantially erode the speedups expected from sparse computation.

It is worth noting that Wanda, RIA, and BaWA all rely on a calibration set to simulate activations in order to obtain the activation scores required for weight pruning, after which a single offline pruning pass is performed. However, these methods have not yet been applied to DiT models in practice. Accordingly, in our implementation, we directly compute activations at each inference step and apply the corresponding scoring rules to perform online pruning of the current weights, aggregating the results over time. **Under this setting, the resulting measurements are, in principle, superior to those obtained by fixing activation statistics from a calibration set, thereby ensuring fairness in comparison.** For hyperparameter configurations, we strictly follow the original papers: in RIA, the exponent $a$ is set to $0.5$; BaWA adopts its default parameters $\theta_1 = 0.5$, $\theta_2 = 0.5$, and $\theta_3 = 1$. For the LoRA fine-tuning stage in SLiM, we adhere to the procedure in the original work: after pruning, we compute the SVD of the difference between the pruned and original weights to initialize the low-rank factors, and set the rank to $0.1 \times \dim$. All remaining hyperparameters and training protocols are kept identical to those used in our LoRA training, ensuring both accuracy and reproducibility of the experimental results.

## F.4. Image Quality Evaluation Metrics

We evaluate image quality from multiple perspectives using four complementary metrics:

- **FID**(Heusel et al., 2017; Parmar et al., 2022) is a distribution-level metric that extracts high-level features with Inception-v3, fits Gaussian distributions to features from real and generated images, and computes the Fréchet distance between them; lower values indicate that the generated distribution more closely matches the real one, jointly reflecting fidelity and diversity.

- **ImageReward**(Xu et al., 2023) is an automated human-preference metric for text-to-image generation that employs a learned reward model to directly output a scalar score for a given text–image pair, approximating human judgments of semantic alignment, visual plausibility, and overall aesthetics; models are typically compared by the average score on a test set.

- **CLIP.iqa**(Wang et al., 2023) is a no-reference image quality assessment method based on CLIP's vision–language representations, which evaluates the perceptual quality of a single image by its relative similarity to textual descriptors such as "high quality" and "low quality," thereby complementing distribution-level metrics in terms of sharpness and naturalness.

- **CLIP.Score**(Hessel et al., 2021) computes the cosine similarity between generated images and text prompts in CLIP's joint embedding space to measure semantic consistency, where higher scores indicate better text–image alignment.

## F.5. Implementation Details

The implementation details have been described earlier. Here, we summarize the experimental setup to facilitate a rapid understanding of the core configurations.

For all models, we fix the random seed to 42 during inference. Unless otherwise specified, all images are generated at a resolution of $1024 \times 1024$. For Qwen-Image, we set `cfg_scale` to 4.0 with 40 inference steps; for FLUX.1-dev, `cfg_scale` is set to 2.0 with 50 steps; and for Z-Image, `cfg_scale` is set to 1.0 with 8 steps. These configurations are applied consistently in all experiments unless otherwise specified.

For the sparsification strategy, we adopt the replacement rule in Algorithm 2 and substitute all image-related layers in the double-stream Qwen-Image models with their corresponding linear layers. The rationale for restricting replacements to the img branch, together with its empirical implications, is discussed in Appendix B. For the double-single-stream FLUX.1-dev model, we similarly replace all img-related layers in the initial double-stream stage; in the subsequent single-stream stage, however, we skip the Out_Down layer to preserve model accuracy. For the single-stream Z-Image-turbo model, we exclude the Out and Up layers, which exhibit large perturbation energy, ensuring that the disturbance introduced by activation sparsification can be effectively compensated by a minimal LoRA. A detailed analysis is provided in Appendix **??**, and all other unspecified Linear layers apply sparse strategies.

During LoRA fine-tuning, we adopt a teacher–student paradigm, where the student is trained to align its outputs with those of the teacher by minimizing their discrepancy. The training procedure and dataset specifications are detailed in Appendix D. We uniformly set the LoRA rank to 64 across all models, striking a stable balance between accuracy and inference efficiency. A detailed analysis is provided in Appendix D.

For orthogonal validation with FP8 quantization, we apply symmetric FP8 quantization (Shen et al., 2024) to activations and directly quantize weights to FP8, evaluating the compatibility of our method with low-precision computation.

For orthogonal validation with distillation, we evaluate on Z-Image-turbo, which has already undergone extreme distillation, to assess compatibility with such strategies. While the original model is difficult to quantize directly to FP8 (Community, 2026), our approach integrates seamlessly with distillation, enabling additional inference acceleration on top of the distillation gains.

# G. Supplemental Results for Accuracy

## G.1. Against SOTA Weight Sparsification

As shown in Tables 12, 13, and 14, our proposed method consistently outperforms current state-of-the-art weight sparsification approaches (e.g., Wanda, RIA, BaWA, and Slim) across multiple architectures, including Qwen-Image, FLUX.1-dev, and Z-Image. We observe that the accuracy of weight-based pruning is highly sensitive to fluctuations in model weights and architectural designs, often leading to significant performance degradation in specific models, such as Z-Image. In contrast, sparsifying activations demonstrates superior robustness and maintains high image quality across all tested scenarios. Further qualitative comparisons are provided in Figures 14, 15, 16, and 17 to visually demonstrate our method's superiority in preserving semantic details and visual fidelity.

*Table 12.* Quantitative image quality comparison of different sparsity strategies on **Qwen-Image-2512**. Arrows in parentheses after each metric indicate the direction of better performance. Bold denotes the best strategy within each row group under the corresponding metric. Besides Full, the table contains two row groups: naive 2:4 sparsity applied to weights and activations, and a comparison between our proposed method and SOTA sparse-weight approaches.

| Method | MJHQ | | | | sDCI | | | |
|---|---|---|---|---|---|---|---|---|
| | FID(↓) | IR(↑) | C.SCR(↑) | C.IQA(↑) | FID(↓) | IR(↑) | C.SCR(↑) | C.IQA(↑) |
| Full | 20.45 | 1.290 | 26.45 | 0.9569 | 24.42 | 1.130 | 25.91 | 0.9435 |
| Sparse Weight | 81.87 | -1.101 | 20.61 | 0.2957 | 119.8 | -1.512 | 20.2760 | 0.2840 |
| Sparse Activation | **39.13** | **0.6575** | **25.60** | **0.7379** | **51.41** | **0.3205** | **26.48** | **0.6593** |
| Wanda (ICLR'24) | 42.21 | 0.2577 | 25.26 | 0.6060 | 59.55 | -0.2371 | 25.54 | 0.5461 |
| RIA (ICLR'24) | 45.36 | 0.1833 | 25.14 | 0.5894 | 62.81 | -0.3143 | 25.59 | 0.5405 |
| BaWA (ICML'25) | 40.53 | 0.2693 | 25.30 | 0.6439 | 56.33 | -0.2107 | 25.68 | 0.5757 |
| Slim (ICML'25) | 27.85 | 1.142 | 26.33 | 0.9369 | 30.22 | 1.103 | 25.83 | 0.9279 |
| Ours | **20.84** | **1.302** | **26.54** | **0.9484** | **24.10** | **1.132** | **26.07** | **0.9285** |

*Table 13.* Quantitative image quality comparison of different sparsity strategies on **FLUX.1-dev**. Arrows in parentheses after each metric indicate the direction of better performance. Bold denotes the best strategy within each row group under the corresponding metric. Besides Full, the table contains two row groups: naive 2:4 sparsity applied to weights and activations, and a comparison between our proposed method and SOTA sparse-weight approaches.

| Method | MJHQ | | | | sDCI | | | |
|---|---|---|---|---|---|---|---|---|
| | FID(↓) | IR(↑) | C.SCR(↑) | C.IQA(↑) | FID(↓) | IR(↑) | C.SCR(↑) | C.IQA(↑) |
| Full | 22.05 | 1.010 | 26.51 | 0.9401 | 25.96 | 1.074 | 26.01 | 0.9487 |
| Sparse Weight | **48.43** | 0.4067 | **25.05** | 0.7695 | 56.86 | 0.5568 | 25.20 | 0.7345 |
| Sparse Activation | 49.87 | **0.4520** | 24.83 | **0.7881** | **52.88** | **0.5900** | **25.74** | **0.7471** |
| Wanda (ICLR'24) | 29.85 | 0.7113 | 25.70 | 0.9042 | 35.51 | 0.7869 | 25.79 | 0.9116 |
| RIA (ICLR'24) | 30.78 | 0.6881 | 25.69 | 0.8938 | 36.48 | 0.7651 | 25.75 | 0.9010 |
| BaWA (ICML'25) | 30.66 | 0.7027 | 25.72 | 0.8998 | 35.97 | 0.7926 | 25.86 | 0.9072 |
| Slim (ICML'25) | 27.06 | 0.7978 | 26.05 | 0.9192 | 31.94 | 0.9315 | **26.21** | 0.9288 |
| Ours | **21.17** | **1.011** | **26.42** | **0.9381** | **24.41** | **1.091** | 25.90 | **0.9445** |

*Table 14.* Quantitative image quality comparison of different sparsity strategies on **Z-Image**. Arrows in parentheses after each metric indicate the direction of better performance. Bold denotes the best strategy within each row group under the corresponding metric. Besides Full, the table contains two row groups: naive 2:4 sparsity applied to weights and activations, and a comparison between our proposed method and SOTA sparse-weight approaches.

| Method | MJHQ | | | | sDCI | | | |
|---|---|---|---|---|---|---|---|---|
| | FID(↓) | IR(↑) | C.SCR(↑) | C.IQA(↑) | FID(↓) | IR(↑) | C.SCR(↑) | C.IQA(↑) |
| Full | 25.70 | 0.9928 | 25.57 | 0.9307 | 25.40 | 0.9974 | 25.91 | 0.9467 |
| Sparse Weight | 360.2 | -2.258 | 13.63 | 0.3492 | 374.2 | -2.2680 | 15.68 | 0.3552 |
| Sparse Activation | **42.65** | **0.5982** | **25.42** | **0.7917** | **36.18** | **0.7306** | **26.72** | **0.7508** |
| Wanda (ICLR'24) | 69.86 | -0.1604 | 23.61 | 0.6224 | 59.98 | -0.0886 | 24.81 | 0.6341 |
| RIA (ICLR'24) | 62.13 | -0.0313 | 23.84 | 0.6613 | 49.69 | 0.0982 | 25.09 | 0.6820 |
| BaWA (ICML'25) | 43.41 | 0.4364 | 24.41 | 0.8221 | 31.89 | 0.5396 | 25.29 | 0.8536 |
| Slim (ICML'25) | 31.37 | 0.7135 | 24.96 | 0.8808 | 27.31 | 0.8457 | 26.19 | 0.8899 |
| Ours | **26.17** | **0.9673** | **25.39** | **0.9250** | **24.93** | **0.9803** | **26.22** | **0.9396** |

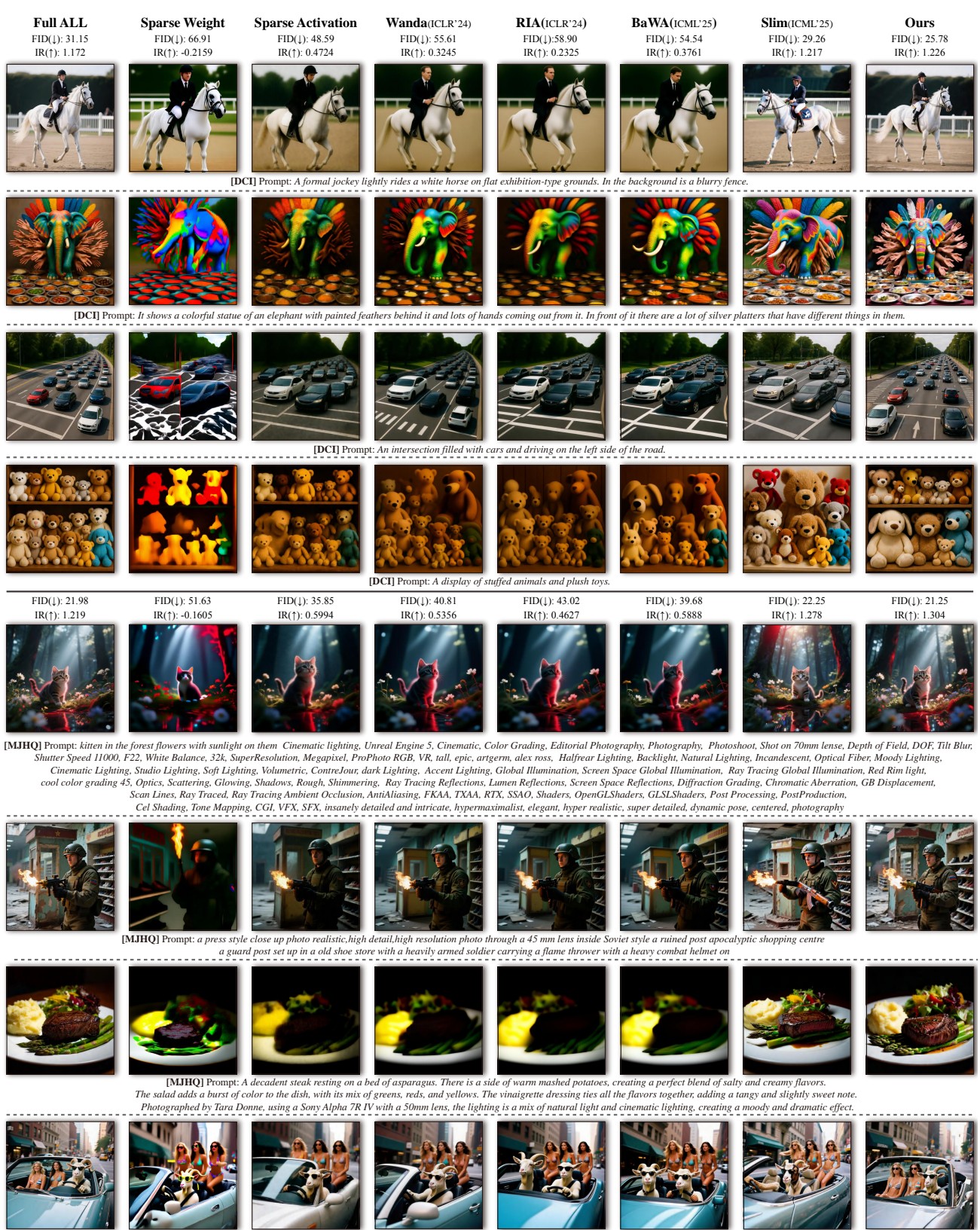

*Figure 14.* Qualitative visual supplement results on **Qwen-Image** with different sparsity strategies over sDCI(DCI) and MJHQ. FID and IR are computed on the full datasets. Our activation sparsity consistently preserves higher visual quality than competing methods.

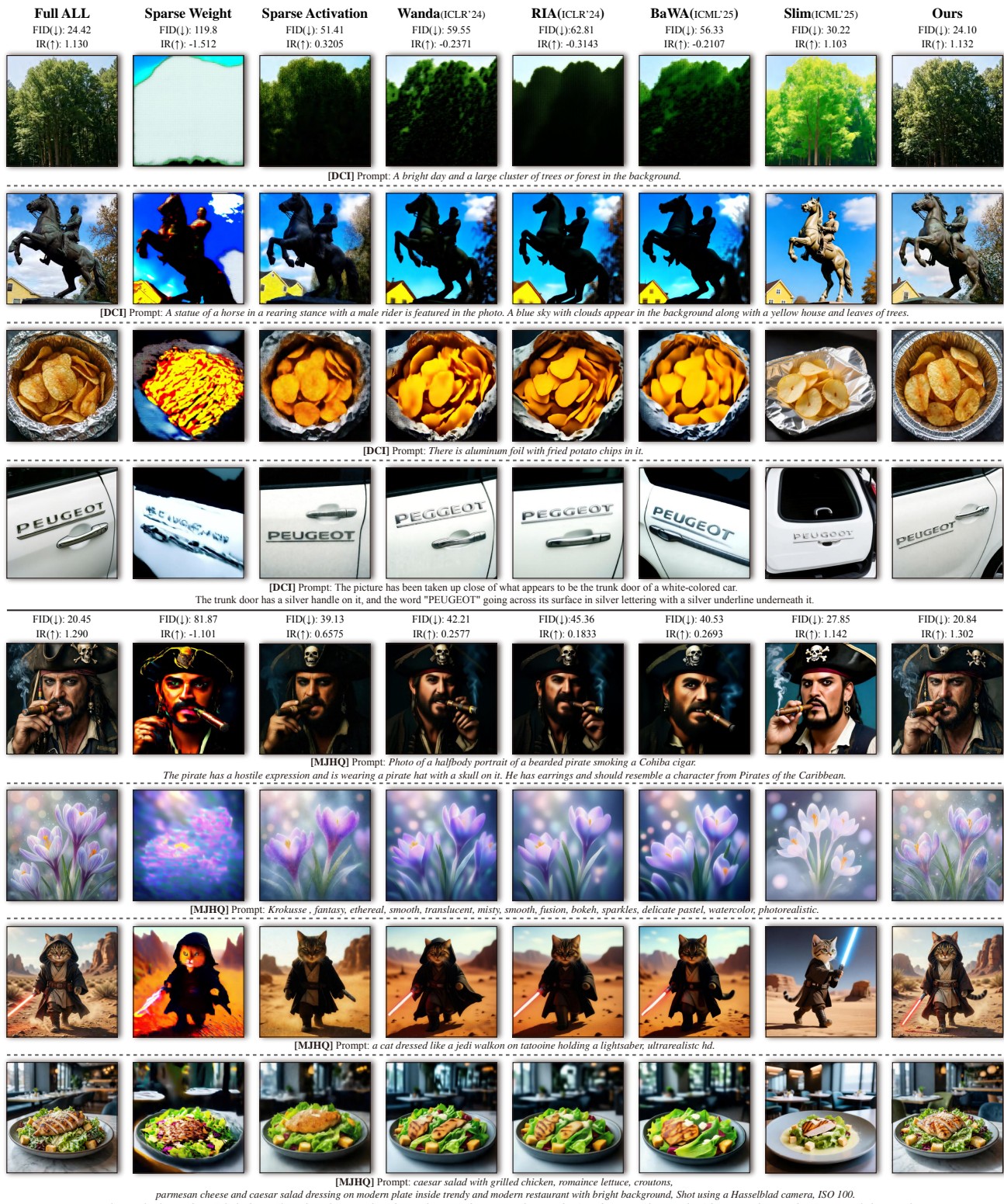

*Figure 15.* Qualitative visual results on **Qwen-Image-2512** with different sparsity strategies over sDCI(DCI) and MJHQ. FID and IR are computed on the full datasets. Our activation sparsity consistently preserves higher visual quality than competing methods.

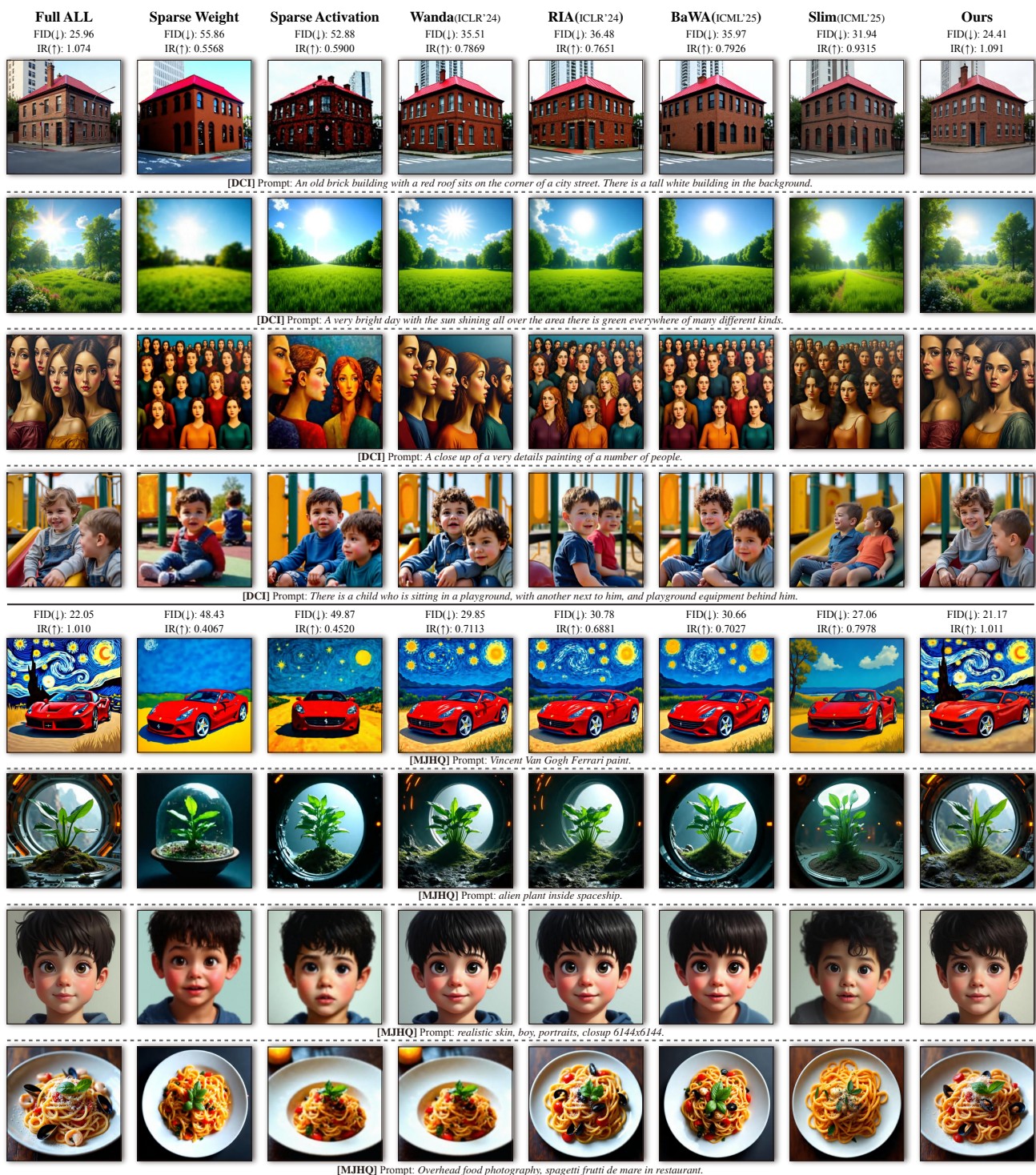

*Figure 16.* Qualitative visual results on **FLUX.1-dev** with different sparsity strategies over sDCI(DCI) and MJHQ. FID and IR are computed on the full datasets. Our activation sparsity consistently preserves higher visual quality than competing methods.

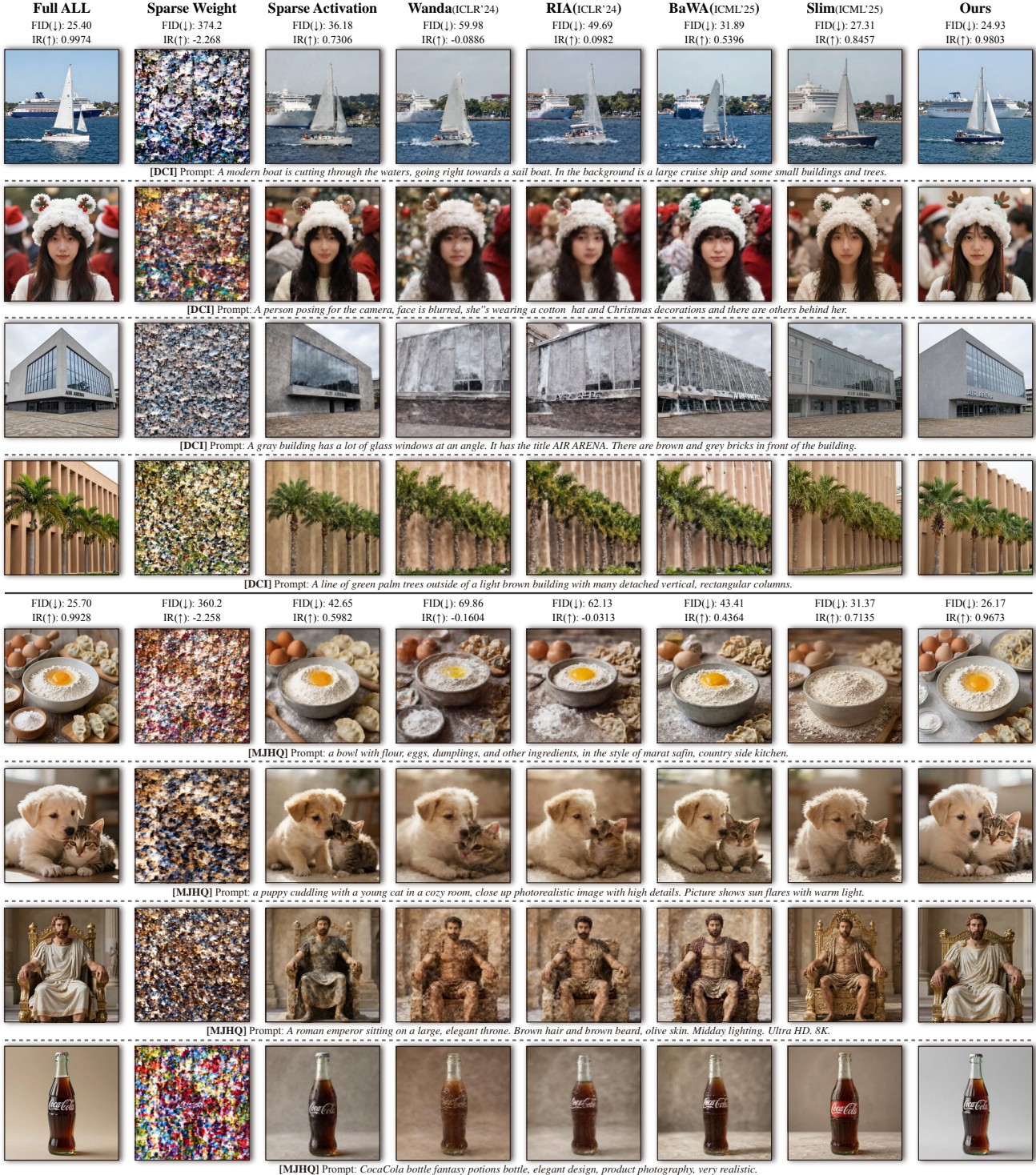

*Figure 17.* Qualitative visual results on **Z-Image** with different sparsity strategies over sDCI(DCI) and MJHQ. FID and IR are computed on the full datasets. Our activation sparsity consistently preserves higher visual quality than competing methods.

## G.2. Ablation

Figure 18 presents additional qualitative ablation results on DCI (sDCI) and MJHQ datasets. The visualizations clearly show that our full configuration—comprising SA, NC, LoRA, and SL—significantly outperforms other variants in preserving semantic details and visual realism.

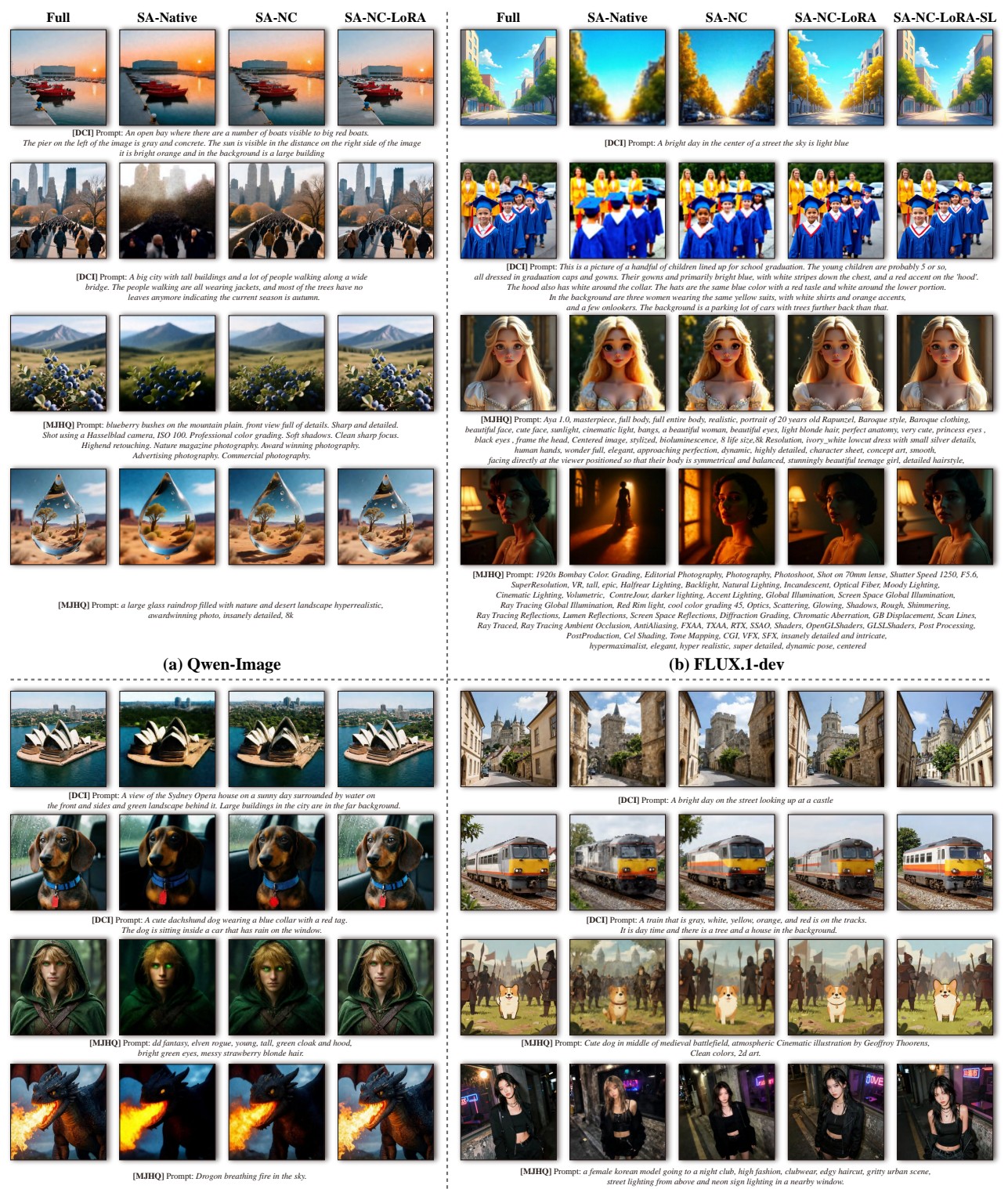

*Figure 18.* Visual ablation results of four categories of models on the DCI(sDCI) and MJHQ datasets; Here, SA denotes Sparse Activation, NC denotes Norm Compensation, LoRA indicates the use of LoRA adaptation, and SL refers to layer skipping in single-stream models.

## H. Supplemental Results for SpeedUp

Table 15 evaluates the efficiency of various GEMM backends on the RTX 4090, demonstrating that our RT-Lynx kernel achieves acceleration patterns nearly identical to those observed on enterprise-grade hardware like the H20. RT-Lynx consistently outperforms PyTorch-SpMM and cuSparseLt, delivering a significant speedup of up to $1.79\times$. Crucially, RT-Lynx drastically minimizes the online sparse overhead (reducing Sparse-Cost to as low as 4.38%), effectively overcoming the bottleneck where metadata processing often offsets the computational gains of N:M sparsity. This robust performance across diverse matrix scales—including those specific to Qwen-Image—establishes a high-efficiency foundation for achieving end-to-end online sparse inference acceleration.

*Table 15.* Performance of Dense and Sparse GEMM Backends on RTX 4090 GPUs. Values show Latency(ms); numbers in parentheses denote speedup over GEMM. PyTorch-SpMM measures the online N:M sparse GEMM time using PyTorch. Sparse-Cost reports the proportion of sparse overhead in the total online sparse execution. Gray rows indicate matrix sizes used in Qwen-Image.

| Matirx Shape | | GEMM | Pytorch-SpMM | | cuSparseLt | | RT-Lynx | |
| --- | --- | --- | --- | --- | --- | --- | --- | --- |
| $M = N$ | $K$ | Latency | Latency | Sparse-Cost | Latency | Sparse-Cost | Latency | Sparse-Cost |
| 2048 | 3072 | 0.1618 | 0.5226 (0.31×) | 38.18% | 0.2408 (0.67×) | 64.04% | 0.1002 (**1.61**×) | **10.97**% |
| 4096 | 3072 | 0.6301 | 0.6053 (1.04×) | 44.31% | 0.5956 (1.06×) | 47.61% | 0.3540 (**1.78**×) | **5.79**% |
| 8192 | 3072 | 2.5850 | 1.8729 (1.38×) | 29.43% | 1.9869 (1.30×) | 30.51% | 1.4466 (**1.79**×) | **4.38**% |
| 2048 | 12288 | 0.6670 | 0.8850 (0.75×) | 63.24% | 0.9122 (0.73×) | 66.72% | 0.4003 (**1.67**×) | **16.64**% |
| 4096 | 12288 | 2.6281 | 2.3781 (1.11×) | 47.85% | 2.5827 (1.02×) | 52.84% | 1.5198 (**1.73**×) | **13.09**% |
| 8192 | 12288 | 10.5636 | 7.4751 (1.41×) | 29.80% | 7.6654 (1.38×) | 35.18% | 6.2600 (**1.69**×) | **6.41**% |

Meanwhile, as shown in the Figure 19, both FLUX.1-dev and Z-Image exhibit behavior consistent with Qwen-Image: activation sparsity yields systematic latency reductions across all major linear components, including MLP projections and attention-related QKV and output layers. The sparse execution path consistently outperforms its dense counterpart at every layer, leading to uniform acceleration throughout the network. This confirms that the proposed strategy generalizes well across different architectures and input aspect ratios, and that sparsity-induced gains are not confined to a single model but are reliably manifested within each computational module of diverse vision Transformers.

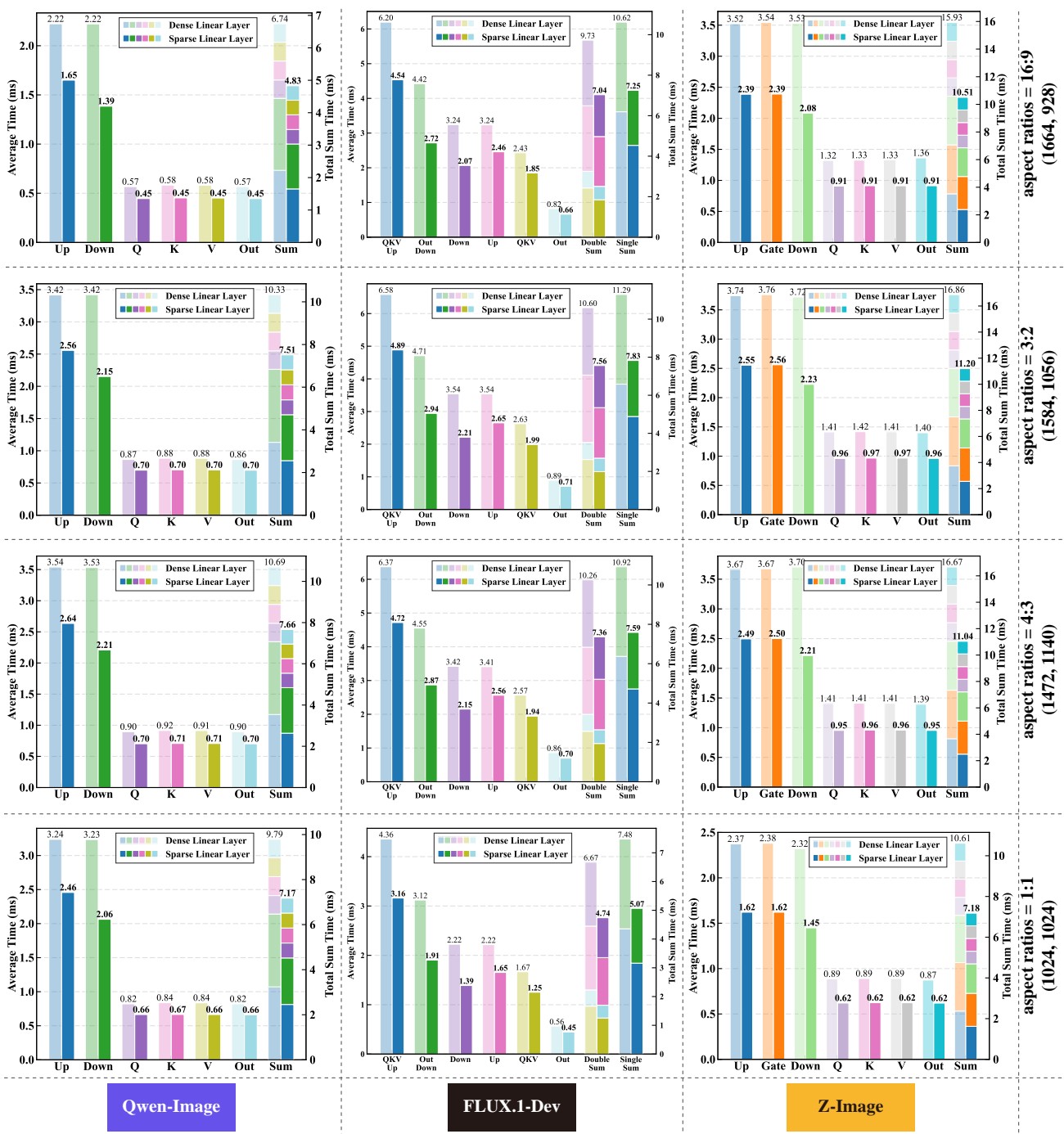

*Figure 19.* Average runtime of dense execution and online activation sparsification in Transformer linear layers across three models for image generation with varying aspect ratios (Qwen-Image: 60 layers; FLUX.1-dev: 18 double-stream layers and 38 single-stream layers; Z-Image: 30 layers).

