# OpenReview forum: "RT-Lynx: Putting the GEMM Sparsity In a Right Way for Diffusion Models"
_ICML.cc/2026/Conference — ICML 2026 regular_

### Official Review · Reviewer_V2tt · 2026-02-26

**Soundness:** 4
**Presentation:** 3
**Significance:** 3
**Originality:** 3
**Overall Recommendation:** 4
**Confidence:** 3

**Summary:**

This paper proposes a novel approach that shifts the sparsification paradigm for Diffusion Transformers from weight sparsity to activation sparsity. The key insight is that DiT activations exhibit intrinsic sparsity due to the superposition mechanism, making them significantly more robust to N:M semi-structured sparsification than weights. Extensive experiments on Qwen-Image, FLUX.1-dev, and Z-Image demonstrate up to 1.88× Sparse GEMM speedup and 1.55× linear-layer speedup with negligible quality degradation.

**Compliance With Llm Reviewing Policy:**

Affirmed.

**Final Justification:**

The author resolved my issue; I am willing to raise score.

**Key Questions For Authors:**

1. You observe that DiT activations exhibit sparsity due to the "superposition mechanism", citing work on LLMs. However, DiTs process image patches rather than discrete tokens in language, and their attention mechanisms operate on spatial rather than sequential structures. Could you provide a more detailed analysis of why DiT activations specifically exhibit this sparsity pattern?

2. Your evaluation covers three DiT models (Qwen-Image, FLUX.1-dev, Z-Image), but does not include larger models like video diffusion models(Wan, HunayunVideo). Would the speedup trends generalize to larger models or video DiTs where activation patterns may differ? Have you conducted any experiments to validate cross-model transfer of sparsification patterns?

**Limitations:**

No. The authors do not adequately discuss the limitations of their work. The paper would benefit from an explicit limitations section addressing:

1. Lack of evaluation on larger video DiTs
2. Absence of comparison with recent caching-based acceleration methods (TeaCache, TaylorSeer, etc.)
3. Theoretical understanding of DiT-specific sparsity mechanisms remains limited

**Strengths And Weaknesses:**

### Strengths

1. The paper identifies a fundamental limitation in prior weight-sparsification approaches and proposes a principled alternative based on activation sparsit.

2. Evaluation covers three major DiT architectures. Visual quality is well-preserved with minimal artifacts compared to weight-sparsification methods

3. Demonstrates orthogonality with existing compression techniques, suggesting practical deployment value

### Weaknesses

1. The core techniques (activation sparsification, norm compensation, LoRA adaptation) have been extensively studied in LLM compression literature. The application of these techniques to DiTs, while valuable, represents an incremental adaptation rather than a fundamental methodological breakthrough.

2. While the paper observes that activations exhibit superposition-induced sparsity, it lacks a rigorous theoretical analysis of why DiT activations specifically exhibit this property. No formal characterization of how the token-level superposition mechanism in DiTs differs from LLMs, and whether this has implications for sparsification strategies.

3. Missing comparison with recent state-of-the-art DiT acceleration methods like TaylorSeer, TeaCache, ProfilingDiT, and SpeCa.

4. The paper reports kernel-level speedups, yet end-to-end speedup is only about 1.2×, indicating significant non-linear-layer overheads. Memory usage implications of the proposed method are not analyzed, particularly for the LoRA compensation branch which requires additional parameter storage.

5. Critical hyperparameters for LoRA training (learning rate, batch size, number of steps) are not provided in the main paper.

---

> ### Author Rebuttal · Authors · 2026-03-31
>
> Dear V2tt:
>
> We sincerely thank you for recognizing the value and experimental completeness of RT-Lynx, as well as for your thoughtful feedback. Below we provide point-by-point responses for your comments and suggestions.
>
> ---
>
> ### **Q1: Significance of Method**
>
> We politely argue that the major contribution of our work is to **propose a paradigm shift in applying sparsification to DiT models**, along with an accuracy-guaranteed activation sparsification algorithm and a highly optimized kernel implementation. Notice that 2:4 sparse has rarely been used in real industrial production scenarios because previous methods only considered using weight sparsification. Our analysis clearly reveals that activation admits a more sparsification-friendly pattern, and we designed norm compensation and LoRA tuning to further reduce the sparsification error. To the best of our knowledge, _this is the first work to successfully achieve lossless acceleration using 2:4 sparsification_. We believe our work can offer a new research perspective and paradigm shift for future studies; therefore, its importance should not be underestimated.
>
> ### **Q2: Origin of Activation Sparsity in DiT**
>
> We do agree that understanding the origin of sparsity in DiTs is an important and interesting topic, and that a theoretical analysis of activation sparsity would be a valuable direction for future research. However, this aspect is not the main focus of our current study. Instead, we focus on sparsifying activations and demonstrate that it enables lossless 2:4 sparsification in practice, providing insights for future theoretical work.
>
> ### **Q3: Relation to Recent DiT Caching Acceleration Methods**
>
> We appreciate the reviewer for this valuable suggestion. While recent methods (e.g., TaylorSeer, TeaCache, ProfilingDiT, SpeCa) focus on step- or feature-level redundancy, RT-Lynx works at the operator/GEMM level, **making them inherently complementary**. In the revision, we will include a discussion and clarify how they relate to our approach.
>
> We further validate this complementarity on FLUX.1-dev (see table below). RT-Lynx composes effectively with TeaCache [1], achieving `3.13×` end-to-end while **maintaining comparable quality**. Additional compatibility with other methods (e.g., sparse attention [2]) is provided in [R2-Q1]() and [R3-Q6]().
>
> |Method|End2End(s)|
> |---|---|
> |Baseline|77.99|
> |*RT-Lynx*|*63.00 (1.24×)*|
> |TeaCache(l=0.6)|29.56 (2.64×)|
> |**RT-Lynx+TeaCache(l=0.6)**|**24.89 (3.13×)**|
>
> |Method|MJHQ FID(↓)|MJHQ IR(↑)|MJHQ C.SCR(↑)|MJHQ C.IQA(↑)|sDCI FID(↓)|sDCI IR(↑)|sDCI C.SCR(↑)|sDCI C.IQA(↑)|
> |---|---|---|---|---|---|---|---|---|
> |Baseline|22.05|1.010|26.51|0.9401|25.96|1.074|26.01|0.9487|
> |*RT-Lynx*|*21.17*|*1.011*|*26.42*|*0.9381*|*24.41*|*1.091*|*25.90*|*0.9445*|
> |TeaCache(l=0.6)|20.67|0.937|25.83|0.9373|25.76|1.024|25.46|0.9487|
> |**RT-Lynx+TeaCache(l=0.6)**|**20.57**|**0.941**|**25.73**|**0.9414**|**24.98**|**1.026**|**25.32**|**0.9443**|
>
> ### **Q4: Memory Overhead in Inference**
>
> We appreciate the reviewer's concern. LoRA introduces only ~0.73% additional parameters in the Qwen-Image model (Appendix D.2), making its **memory overhead negligible**. Peak memory increases are also minimal:
>
> - Qwen-Image: 57239.50 → 57673.28 MB (**0.76%**)
> - FLUX.1-dev: 34464.07 → 34778.57 MB (**0.92%**)
> - Z-Image: 21934.57 → 22164.63 MB (**1.05%**)
>
> This is due to our fused execution design that avoids extra intermediate tensors. Overall, RT-Lynx incurs only marginal and well-controlled memory overhead.
>
> ### **Q5: LoRA Training Details**
>
> _The LoRA training configuration is not missing_; full details (e.g., learning rate, batch size, training steps) are provided in Appendix D.1 and F.5. We did not elaborate on it in the main text due to space constraints. In the revision, we will include a brief summary in the main paper and clearly refer readers to the appendix for complete settings.
>
> ### **Q6: Extension to Video Models**
>
> We sincerely thank the reviewer for this thoughtful question. For a detailed discussion, we kindly invite the reviewer to refer to our response to [R1-Q5](), where we present preliminary analyses and experiments on video DiTs (e.g., Wan and LTX). In brief, although full-scale evaluation on video models remains future work due to their high computational cost, our preliminary analyses on video DiTs show consistent trends: **activation sparsity consistently induces lower error than weight sparsity**.
>
> ### **Q7: Explicit Limitations**
>
> We will add a dedicated limitations section in the revision. Regarding composability, RT-Lynx is orthogonal to step-level methods (e.g., TeaCache, TaylorSeer), and we will include additional results on their combination.
>
> ---
>
> ### **References**
>
> [1] Timestep Embedding Tells: It's Time to Cache for Video Diffusion Model
>
> [2] SpargeAttention: Accurate and Training-free Sparse Attention Accelerating Any Model Inference

---

> > ### Author Rebuttal · Reviewer_V2tt · 2026-04-03
> >
> > The author resolved my issue; I am willing to raise score.

---

> > > ### Author Response · Authors · 2026-04-03
> > >
> > > Dear V2tt:
> > >
> > > Thank you for your positive feedback and for the time and effort you invested in reviewing our work. We will further refine the manuscript to better present our activation-based sparsification for practical lossless 2:4 acceleration in DiT models.
> > >
> > > Best regards, The Authors

---

### Official Review · Reviewer_HpCM · 2026-02-28

**Soundness:** 3
**Presentation:** 3
**Significance:** 3
**Originality:** 3
**Overall Recommendation:** 5
**Confidence:** 4

**Summary:**

Diffusion Transformers (DiTs) achieve remarkable high-quality image generation but suffer from severe inference efficiency bottlenecks due to compute-intensive iterative steps. The authors investigate semi-structured 2:4 sparsity to accelerate DiTs. They discover that traditional weight sparsification degrades generation quality because weights follow a quasi-Gaussian distribution and are not intrinsically sparse. In contrast, DiT activations are highly sparse due to token-level superposition, with only 5% to 10% of neurons being active. The paper proposes RT-Lynx, a framework that shifts the paradigm from weight sparsification to activation sparsification. RT-Lynx features a norm-compensated activation scheme to preserve the $l_{2}$ norm of original activations, a lightweight LoRA branch to recover residual fine-grained details, and selective layer skipping for single-stream paths. To combat the high overhead of online activation sparsification (which can natively occupy up to 59% of runtime), the authors designed a fused CUDA execution pipeline that integrates online pattern determination, Top-K selection, and compression directly into Sparse Tensor Core compatible layouts. RT-Lynx achieves up to 1.55x speedups in linear layers and ~1.2x end-to-end acceleration. It substantially outperforms state-of-the-art weight sparsification methods (like Wanda, RIA, and BaWA) in generation quality across multiple architectures (Qwen-Image, FLUX.1-dev, and Z-Image).

**Compliance With Llm Reviewing Policy:**

Affirmed.

**Key Questions For Authors:**

1. Could you clarify how sensitive the LoRA fine-tuning phase is to the diversity and quality of the generated prompt-image pairs used for training?

2. Is there a systematic or automated thresholding method based on the Relative Frobenius Error (RFE) to determine exactly which layers to skip in new, unseen single-stream architectures, or must this be manually profiled per model? In addition, it is unclear why relative Frobenius error is adopted as the primary criterion for analysis. It would be helpful if the authors could elaborate on the specific motivations behind this choice and clarify whether there are particular theoretical or empirical considerations that justify using RFE over alternative metrics.

3. While the paper excels in latency analysis, could you provide a brief clarification on the peak memory overhead during inference when both the sparse main path and the dense LoRA branch are executed concurrently?

4. CUDA kernel optimization is a key component of the method, yet the implementation details are limited. Including pseudocode or a clearer algorithmic description would improve transparency and reproducibility.

5. The Related Work section on diffusion acceleration is not sufficiently comprehensive, as several representative works on sparse attention and cache-based acceleration are not discussed.

    [1] DiTFastAttn: Attention Compression for Diffusion Transformer Models

    [2] DiTFastAttnV2: Head-wise Attention Compression for Multi-Modality Diffusion Transformers

    [3] Sparse VideoGen: Accelerating Video Diffusion Transformers with Spatial-Temporal Sparsity

    [4] DeepCache: Accelerating Diffusion Models for Free

    [5] FORA: Fast-Forward Caching in Diffusion Transformer Acceleration

    [6] Δ\-DiT: A Training-Free Acceleration Method Tailored for Diffusion Transformers

    Moreover, although the authors claim that RT-Lynx is orthogonal to existing acceleration methods, no empirical evidence is provided. Given that the reported end-to-end speedup (~1.2×) is relatively modest, demonstrating scalability and compatibility with other techniques would be important.

**Limitations:**

Yes. The authors have adequately discussed the limitations and potential negative societal impact of their work.

**Strengths And Weaknesses:**

### Paper Strengths

- The empirical analysis distinguishing the sparsity distributions of weights versus activations is compelling. This justifies the core paradigm shift of the paper and provides a strong foundation for the proposed method.

- The paper presents a comprehensive solution that bridges algorithmic design with low-level systems engineering. Fusing the online sparsification pipeline into a single CUDA execution path effectively solves the practical latency bottlenecks of dynamic sparsity.

- The proposed method demonstrates near-lossless generation quality, significantly outperforming recent training-free weight sparsification baselines on established metrics (FID, Image Reward, CLIP-IQA).

- The authors successfully demonstrate that RT-Lynx is orthogonal to standard acceleration techniques, showing near-lossless compatibility with W8A8 quantization and extreme step distillation.


### Paper Weaknesses

- Unlike zero-shot or one-shot calibration methods (like Wanda or RIA), RT-Lynx requires a LoRA fine-tuning phase of roughly 2k steps over a generated dataset to recover performance. While the overhead is low, it makes the method less "plug-and-play" than purely training-free alternatives.

- The strategy of "selective layer skipping" for single-stream models (e.g., skipping `attn.o_proj` and `mlp.up` in Z-Image) appears somewhat empirical. It relies on measuring the Relative Frobenius Error (RFE) across layer types, which may require manual profiling and trial-and-error when adapting the framework to novel architectures.

- The reported computational speedup appears to be closely tied to NVIDIA’s Sparse Tensor Cores (SpTC) and the specific 2:4 structured sparsity pattern. However, the paper does not seem to provide experimental results under alternative sparsity configurations, such as 1:2 or 4:8 patterns.

---

> ### Author Rebuttal · Authors · 2026-03-31
>
> Dear HpCM:
>
> We sincerely appreciate your support and thoughtful feedback. Below we provide point-by-point responses for your comments and suggestions.
>
> ---
>
> ### **Q1: The Design of Layer Skipping**
>
> The choice for layer skipping is based on _RFE_ (see our discussion in Appendix D), where we perform a single forward pass to profile _RFE_ and use a simple Top-K rule to skip high-sensitivity layers (Fig.13). However, we do agree that a fine-grained and automatic design for layer skipping is also an important extension to RT-Lynx. We will add a discussion part for this in our revised version.
>
> ### **Q2: Alternative Sparse Configuration**
>
> We adopt 2:4 semi-structured sparsity because it is the only pattern natively supported by NVIDIA Sparse Tensor Cores and provides stable acceleration across `FP16/BF16/FP8/INT8` (Table 5). In contrast, 1:2 is limited to TF32, and 4:8 mainly targets low-precision settings (e.g., INT4/FP4), which are not aligned with standard DiT inference (FP16/BF16), making fair comparison difficult. That said, **our method is not tied to a specific sparsity ratio**. It is built on a general framework of activation sparsity with fused online execution.
>
> ### **Q3: Sensitivity of LoRA**
>
> We politely point out that our objective is to _align the sparse model with the full model's outputs_ (not real image), which mitigates overfitting in LoRA training and makes RT-Lynx robust to general prompt/image datasets. While higher-quality prompts may yield marginal gains, under the same LoRA budget, RT-Lynx consistently outperforms weight sparsity with LoRA compensation (see Tables 1, 10, 11, and 12), underscoring its advantage.
>
> ### **Q4: Memory Overhead in Inference**
>
> LoRA introduces minimal parameters (Appendix D.2), with only `~0.73%` overhead at rank 64 in the Qwen-Image model, making its parameter-level memory cost negligible. We further report _peak memory comparisons_ in the inference stage:
>
> - Qwen-Image: 57239.50 → 57673.28 MB (**0.76%**)
> - FLUX.1-dev: 34464.07 → 34778.57 MB (**0.92%**)
> - Z-Image: 21934.57 → 22164.63 MB (**1.05%**)
>
> These results show that RT-Lynx introduces only a slight increase in memory, consistent with expectations. This is enabled by our fused execution design, which avoids extra intermediate materialization. Overall, _RT-Lynx maintains acceleration with minimal and well-controlled memory overhead_.
>
> ### **Q5: CUDA Kernel Details**
>
> We present the overall workflow in Alg. 1 and the fused pipeline design in Fig.5. We agree that additional implementation details would improve clarity, and we will add detailed pseudocode to the appendix in the revision. We will also **open-source the full implementation** upon acceptance.
>
> ### **Q6: Insufficient Related Works and Orthogonality**
>
> We thank the reviewer for this valuable suggestion. In the revision, we will include the works mentioned (e.g., DiTFastAttn, Sparse VideoGen, DeepCache) and clarify their relationship to our method. In addition, it should be noted that RT-Lynx focuses on GEMM-level sparsity, which is **totally compatible** with the methods above.
>
> Below, we present additional experimental results demonstrating RT-Lynx's compatibility with other acceleration methods (quantitative compatibility is reported in Table 1). Please refer to our response in [R2-Q1]() for speedup results. Building on these results, we observe that RT-Lynx composes effectively with step distillation, quantization, caching [1], and attention optimization [2], consistently delivering additional speedups while maintaining comparable quality.
>
> |Optimization|Method|MJHQ FID(↓)|MJHQ IR(↑)|MJHQ C.SCR(↑)|MJHQ C.IQA(↑)|sDCI FID(↓)|sDCI IR(↑)|sDCI C.SCR(↑)|sDCI C.IQA(↑)|
> |---|---|---|---|---|---|---|---|---|---|
> |Distillation (Z-Image)|Baseline (step-50)|22.31|0.8790|26.25|0.9046|18.60|0.9545|26.23|0.9196|
> ||*RT-Lynx*|*22.68*|*0.9301*|*26.25*|*0.9058*|*18.80*|*1.006*|*26.25*|*0.9167*|
> ||Turbo (step-8)|25.70|0.9928|25.57|0.9307|25.40|0.9974|25.91|0.9467|
> ||**RT-Lynx+Turbo**|**26.17**|**0.9673**|**25.39**|**0.9250**|**24.93**|**0.9803**|**26.22**|**0.9396**|
> |Cache (FLUX.1-dev)|Baseline|22.05|1.010|26.51|0.9401|25.96|1.074|26.01|0.9487|
> ||*RT-Lynx*|*21.17*|*1.011*|*26.42*|*0.9381*|*24.41*|*1.091*|*25.90*|*0.9445*|
> ||TeaCache(l=0.6)|20.67|0.937|25.83|0.9373|25.76|1.024|25.46|0.9487|
> ||**RT-Lynx+TeaCache(l=0.6)**|**20.57**|**0.941**|**25.73**|**0.9414**|**24.98**|**1.026**|**25.32**|**0.9443**|
> |Attention (Qwen-Image-2512)|Baseline|20.45|1.290|26.45|0.9569|24.42|1.130|25.91|0.9435|
> ||*RT-Lynx*|*20.84*|*1.302*|*26.54*|*0.9484*|*24.10*|*1.132*|*26.07*|*0.9285*|
> ||SpargeAttn(k=0.5)|21.17|1.192|26.18|0.9557|25.46|1.067|26.16|0.9390|
> ||**RT-Lynx+SpargeAttn(k=0.5)**|**21.39**|**1.212**|**26.37**|**0.9420**|**25.22**|**1.075**|**26.42**|**0.9267**|
>
> ---
>
> ### **References**
>
> [1] Timestep Embedding Tells: It's Time to Cache for Video Diffusion Model
>
> [2] SpargeAttention: Accurate and Training-free Sparse Attention Accelerating Any Model Inference

---

> > ### Author Rebuttal · Reviewer_HpCM · 2026-04-04
> >
> > My concerns are resolved. I will maintain my rating as accept.

---

> > > ### Author Response · Authors · 2026-04-04
> > >
> > > Dear HpCM:
> > >
> > > We thank you for your recognition of RT-Lynx and for the time and effort devoted to reviewing our work. We are glad that our revisions have addressed your concerns, and we will incorporate these clarifications to further improve the manuscript.
> > >
> > > Best regards, The Authors

---

### Official Review · Reviewer_CpjC · 2026-03-04

**Soundness:** 4
**Presentation:** 4
**Significance:** 3
**Originality:** 3
**Overall Recommendation:** 5
**Confidence:** 4

**Summary:**

While N:M sparsity (e.g., 2:4 sparsity) has been widely explored for weight pruning in large language models, prior work shows that directly sparsifying weights often causes substantial quality degradation in generative models. The authors instead proposes RT-Lynx, a framework that performs online activation sparsification under a 2:4 pattern and compensates for the resulting approximation error using two mechanisms: (1) a norm-compensation scheme that preserves activation magnitude after sparsification, and (2) a lightweight LoRA branch that learns to recover residual information lost due to sparsity. The paper further introduces an optimized CUDA kernel that fuses sparsification, compression, and sparse GEMM execution to reduce the overhead of online sparsity generation. Experiments across several diffusion models (including Qwen-Image, FLUX, and Z-Image) demonstrate that the method preserves image generation quality while achieving meaningful kernel-level acceleration and moderate end-to-end inference speedups. The method is also shown to be compatible with quantization and distillation pipelines

**Compliance With Llm Reviewing Policy:**

Affirmed.

**Final Justification:**

The paper is technically sound and clearly presented. While the end-to-end speedup is limited by non-linear and attention components, this is an expected systems constraint and does not undermine the core contribution. The rebuttal addressed my concerns, particularly by showing strong composability with other acceleration methods and clarifying deployment details. I maintain my accept recommendation.

**Key Questions For Authors:**

1. How many GPUs are used for lora finetuning?
2. Would this the kernel be compatible with other hardware such as A100 and B200?

**Limitations:**

This paper contains no Impact Statement section before reference. Should it be desk-rejected?

**Strengths And Weaknesses:**

**Strengths**

Soundness:
In my opnion, the paper is very sound. The core motivationthat activations in transformer-based diffusion models exhibit intrinsic sparsity is supported by empirical analysis of activation distributions and error metrics across layers. The proposed solution combines several components (activation sparsification, norm compensation, LoRA recovery, and kernel optimization) in a coherent pipeline. Experimental evaluation is thorough and includes multiple diffusion architectures, strong ablation studies, and comparisons with several existing sparsification approaches such as Wanda, RIA, and BaWA.

Presentation: The paper is clearly written and well structured. I especially appreciate the figures illustrating activation distributions and error comparisons. In addition, the appendix contains substantial implementation details and experimental configurations, which should help reproducibility. The discussion of system-level design, particularly the fused sparse kernel for reducing online sparsification overhead, is also informative.

Significance: The use of 2:4 semi-structured sparsity is appealing because it is already supported by NVIDIA hardware but has seen limited adoption in diffusion models. Demonstrating that activation sparsity can leverage this hardware capability while preserving generation quality is therefore extremely practically relevant. The work may encourage broader exploration of hardware-friendly sparsity patterns in generative models.

Originality: The paper provides a novel perspective by identifying that activation sparsity is significantly more robust than weight sparsity in diffusion transformers. The system-level contribution, namely the fused kernel for online sparsification and sparse GEMM execution, also demonstrates meaningful engineering novelty.

**Weakness**

Significance:  While the method achieves substantial acceleration for sparse GEMM operations, the overall generation pipeline includes many other components (e.g., attention, sampling steps, and non-linear operations), which limits the total latency improvement. However, this limitation is somewhat expected, as the proportion of runtime spent in linear layers varies across models and implementations.

---

> ### Author Rebuttal · Authors · 2026-03-31
>
> Dear CpjC:
>
> We sincerely thank you for the careful reading and highly positive evaluation of our work. Below we provide point-by-point responses for your comments and suggestions.
>
> ---
>
> ### **Q1: End-to-End Speedup Limitation**
>
> Thank you for your insightful observation. RT-Lynx is designed to accelerate linear layers in DiTs and, importantly, is **fully complementary to other inference optimizations** such as step distillation, quantization, caching [1], and attention acceleration [2]. This enables it to integrate seamlessly into existing pipelines.
>
> To substantiate its practical effectiveness and composability, we report results across multiple acceleration methods (see table below). RT-Lynx consistently composes with these methods to deliver additional speedups. This composability makes RT-Lynx a practical and effective module for further improving end-to-end efficiency when combined with a range of acceleration strategies.
>
> |Setting|Method|End2End(s)|
> |---|---|---|
> |Distillation (Z-Image)|Baseline (step-50)|49.44|
> ||*RT-Lynx*|*40.76 (1.21×)*|
> ||Turbo (step-8)|4.99 (9.91×)|
> ||**RT-Lynx+Turbo**|**4.17 (11.86×)**|
> |Quant (Qwen-Image)|Baseline|60.66|
> ||*RT-Lynx*|*50.60 (1.20×)*|
> ||W8A8|54.45 (1.11×)|
> ||**RT-Lynx+W8A8**|**46.04 (1.32×)**|
> |Cache (FLUX.1-dev)|Baseline|77.99|
> ||*RT-Lynx*|*63.00 (1.24×)*|
> ||TeaCache(l=0.6)|29.56 (2.64×)|
> ||**RT-Lynx+TeaCache(l=0.6)**|**24.89 (3.13×)**|
> |Attention (Qwen-Image-2512)|Baseline|60.42|
> ||*RT-Lynx*|*49.35 (1.22×)*|
> ||SpargeAttn(k=0.5)|54.49 (1.11×)|
> ||**RT-Lynx+SpargeAttn(k=0.5)**|**44.35 (1.36×)**|
>
> ### **Q2: LoRA Finetuning Details**
>
> LoRA fine-tuning for all three DiT architectures is conducted on **a single NVIDIA H20 GPU**. Due to the low-rank (e.g., 64) design, LoRA introduces only `~0.73%` additional parameters in the Qwen-Image model, resulting in minimal and manageable memory overhead.
>
> ### **Q3: Hardware Compatibility**
>
> RT-Lynx is built on NVIDIA’s `N:M semi-structured` sparse Tensor Cores and is directly compatible with GPUs that support this feature, including the A100 (Ampere) and newer architectures such as Hopper and Blackwell, delivering up to 2× theoretical speedup. Our fused kernel is implemented using CUDA and CUTLASS, without relying on architecture-specific instructions. This is validated in our experiments: as shown in Table 4 (Hopper) and Table 13 (Ada Lovelace, similar to Ampere), the kernel achieves up to `1.88×` and `1.79×` speedup, respectively. We will **fully open-source** our kernels, inference, and training code to facilitate reproducibility and deployment across different hardware platforms.
>
> ### **Q4: Impact Statement Missing**
>
> We thank the reviewer for the careful suggestion. We will add an Impact Statement in the revision. While our work focuses on inference acceleration, we note that improved efficiency may lower barriers to misuse and recommend appropriate safety measures for deployment.
>
> ---
>
> ### **References**
>
> [1] Timestep Embedding Tells: It's Time to Cache for Video Diffusion Model
>
> [2] SpargeAttention: Accurate and Training-free Sparse Attention Accelerating Any Model Inference

---

> > ### Author Rebuttal · Reviewer_CpjC · 2026-04-01
> >
> > My concerns are fully resolved. I will maintain my rating as "accept".

---

> > > ### Author Response · Authors · 2026-04-01
> > >
> > > Dear CpjC:
> > >
> > > We sincerely appreciate your positive assessment and the time and effort you invested in reviewing our paper. We're pleased that our revisions addressed your concerns and will include these discussions to further strengthen the manuscript.
> > >
> > > Best regards, The Authors

---

### Official Review · Reviewer_gtAL · 2026-03-11

**Soundness:** 3
**Presentation:** 3
**Significance:** 3
**Originality:** 2
**Overall Recommendation:** 4
**Confidence:** 3

**Summary:**

This paper proposes an acceleration method for DiT models based on activation sparsity. It introduces multiple modules, including LoRA for high-frequency detail compensation, norm scaling, and an efficient CUDA kernel implementation. The method demonstrates strong acceleration and quality preservation across multiple models (e.g., Z-Image, Qwen-Image) and various baselines. This paper strikes a good balance between engineering practicality and exploratory innovation.

**Compliance With Llm Reviewing Policy:**

Affirmed.

**Final Justification:**

In the rebuttal with the authors, some of my concerns were addressed (e.g., memory consumption of distillation process, etc.). I believe the experiments are sufficient and have strong engineering significance. However, I still have doubts about how the quality of training prompts affects the final model performance — if this cannot be aligned with other methods, it may lead to unfair comparisons. Based on this, I tend to keep my score, which is weak accept.

**Key Questions For Authors:**

1.In Figure 6, the showcase demonstrate that only RT-Lynx appears to fully preserve the layout and details of the "Full All" baseline. The visual results are quite impressive. I was wondering if the authors could also present and analyze some bad cases of RT-Lynx? I believe this would help further understand the effectiveness and limitations of the proposed method.

2.Is RT-Lynx also effective for video diffusion models, such as Wan or Hunyuan? Has its applicability been explored in the temporal domain?

3.Can the intrinsic sparsity of activations be maintained in deeper layers? Is it possible that shallow blocks are more suitable for activation sparsification, while deeper blocks are not? In other words, does the sparsity pattern degrade across layers?

**Limitations:**

Yes, this paper does not have a standalone "Limitation" section, but its limitations are discussed across multiple chapters.

**Strengths And Weaknesses:**

**Strengths**:

1.The experiments are exceptionally solid. The main experiments, ablation studies, and baseline comparisons are all extremely comprehensive. The results are convincing.

2.This paper demonstrates strong engineering practicality. It leverages the sparsity of activations and successfully makes it work under the 2:4 sparsity pattern, representing a blend of engineering and innovation. Previous work on activation sparsity either could not adapt to hardware or was limited to FFN layers.

3.RT-Lynx can be integrated with techniques like quantization and distillation, which enhances its applicability.

---

**Weakness**:

1.The addition of the LoRA branch explicitly introduces extra parameters (even though the authors conducted experiments in Table 7 showing that a rank of 64 works well). Does this raise concerns about an unfair comparison? For instance, the paper reports that RT-Lynx outperforms the original model, which might also be related to the quality of the dataset (e.g., high-quality prompts).

2.The training dataset for RT-Lynx is derived from a frozen teacher model. Does this imply that training requires significantly more memory? Or is a teacher model not necessary for activation sparsification? Does the paper provide details on this aspect? If only the additional LoRA parameters are involved, the memory overhead would be acceptable.

3.The discussion on how LoRA specifically compensates for high-frequency details is insufficient, as is the explanation for why certain components (e.g. o_proj).

---

> ### Author Rebuttal · Authors · 2026-03-31
>
> Dear gtAL:
>
> We sincerely appreciate your supportive and thoughtful feedback. We understand your concerns regarding fairness, training overhead, and technical clarity, and address them in detail below.
>
> ---
>
> ### **Q1: Fairness after the Introduction of LoRA**
>
> We politely point out that in LoRA training, we only simulate the teacher model's outputs (not the actual images), and we agree that a high-quality prompt might yield some gain for our model. However, under the same LoRA training budget, RT-Lynx greatly outperforms weight-sparse methods using LoRA compensation (see Tables 1, 10, 11, and 12), demonstrating the superiority of our method. We will clarify this in our revised version.
>
> ### **Q2: Training Cost**
>
> The training memory is identical to that of standard LoRA training, since LoRA is the only additional component in RT-Lynx. With `PEFT`, the model can be switched via adapter context, **making the memory overhead negligible**. The teacher model is only used during training and is not involved in inference; only the LoRA parameters are active during inference. We will clarify this in the revision.
>
> ### **Q3: LoRA Compensation Mechanism and Layer-skipping**
>
> Our empirical results show that **low-magnitude activations primarily contribute to high-frequency details** (e.g., hair and textures); removing them would degrade the generated images. The key insight behind this LoRA compensation is that we believe these high-frequency components constitute only a small portion of the image information; therefore, they can be recovered by a LoRA branch. Experimental results also verify this assumption (see Fig. 7 and Fig. 18 for illustration). Additionally, certain layers are skipped because they exhibit significantly higher sparsification error, meaning the information loss cannot be recovered by the LoRA branch. We will elaborate on this content in the revised version.
>
> ### **Q4: Bad Cases**
>
> RT-Lynx may show performance fluctuations on prompts where the base model is already unstable (e.g., complex high-frequency textures or fine structures). As shown in Fig.14 (second row), sparsification errors can be amplified in such cases, leading to noticeable differences. These cases are rare in practice, and we will include additional examples and analysis in the revision.
>
> ### **Q5: Extension to Video Models**
>
> Due to limited time and the high cost of video generation, we have not yet conducted full evaluations on video DiTs. However, we performed preliminary analyses on Wan [1] and LTX [2] following the same motivation. Specifically, we compute the Relative Frobenius Error (_RFE_) introduced by both sparsification strategies, following the same protocol as in Fig. 2 and Fig. 11 (c) (see the table). Results show that **activation sparsity consistently induces significantly lower error** than weight sparsity in both video models. This suggests that activation sparsity is also prominent in video DiTs, supporting the generality of sparsity patterns driven by the superposition mechanism.
>
> |Model|Method|Q|K|V|Out|Up|Down|
> |---|---|---|---|---|---|---|---|
> |Wan2.2-T2V|Sparse Weight|0.2911|0.1578|0.2113|0.1981|0.1831|0.2107|
> ||**Sparse Activation**|**0.1232**|**0.0762**|**0.1050**|**0.0768**|**0.1384**|**0.1709**|
> |LTX2.3-T2AV|Sparse Weight|0.3030|0.1579|0.2090|0.1945|0.1842|0.2075|
> ||**Sparse Activation**|**0.1359**|**0.0793**|**0.1079**|**0.0741**|**0.1324**|**0.1702**|
>
> ### **Q6: Layer-wise Sparsity**
>
> As shown in Fig.2(c) and Fig.11(c), activation sparsity consistently results in much lower _RFE_ than weight sparsity across all layers. We do notice that shallow layers exhibit slightly lower error; beyond that, the error quickly stabilizes and remains consistent across different depths and layer types. This suggests that sparsification does not worsen with depth, and deeper layers remain equally suitable.
>
> ---
>
> ### **References**
>
> [1] Wan: Open and Advanced Large-Scale Video Generative Models
>
> [2] LTX-2: Efficient Joint Audio-Visual Foundation Model

---

> > ### Author Rebuttal · Reviewer_gtAL · 2026-04-01
> >
> > Thank you. This has resolved the majority of my concerns.
> >
> > With respect to Q1, for distillation methods (e.g., DMD), the quality of the prompt itself also significantly influences the final performance of the student model. I am curious about how the training prompts were selected in your work. Additionally, adopting teacher distillation entails greater GPU memory consumption—does this align with the use of LoRA in comparable approaches?

---

> > > ### Author Response · Authors · 2026-04-01
> > >
> > > Dear gtAL:
> > >
> > > We apologize for the ambiguity: **“distillation” here refers to self-distillation**, i.e., aligning the sparse model with the original model’s outputs, rather than step distillation (e.g., DMD) that reduces diffusion steps. Since our objective is output alignment rather than fitting real data distributions, the impact of prompt quality is limited. We use prompts automatically generated by Qwen3 to promote diversity and will include additional experiments with open-source prompt datasets in the revision.
> > >
> > > In terms of training, our method is comparable to standard LoRA training; with `PEFT`, the teacher and sparse model are switched via adapter context, incurring negligible training overhead.

---

### Decision · Program_Chairs · 2026-04-30

**Decision:**

Accept (regular)

**Comment:**

The paper presents an activation sparsity-based acceleration method for DiT models. The rebuttal addressed key concerns, including memory usage, composability, and deployment details. One remaining issue is the impact of training prompt quality on fairness of comparisons. Overall, this concern does not undermine the main contribution, and the paper is a practical and meaningful addition.